# The mitochondrial inner membrane protein LETM1 modulates cristae organization through its LETM domain

Seiko Nakamura[1,8], Aiko Matsui[2,8], Shiori Akabane[2,8], Yasushi Tamura [3], Azumi Hatano[2], Yuriko Miyano[2], Hiroshi Omote[4], Mizuho Kajikawa[5], Katsumi Maenaka[6], Yoshinori Moriyama[4], Toshiya Endo[7] & Toshihiko Oka [2✉]

LETM1 is a mitochondrial inner membrane protein that is required for maintaining the mitochondrial morphology and cristae structures, and regulates mitochondrial ion homeostasis. Here we report a role of LETM1 in the organization of cristae structures. We identified four amino acid residues of human LETM1 that are crucial for complementation of the growth deficiency caused by gene deletion of a yeast LETM1 orthologue. Substituting amino acid residues with alanine disrupts the correct assembly of a protein complex containing LETM1 and prevents changes in the mitochondrial morphology induced by exogenous LETM1 expression. Moreover, the LETM1 protein changes the shapes of the membranes of in vitro-reconstituted proteoliposomes, leading to the formation of invaginated membrane structures on artificial liposomes. LETM1 mutant proteins with alanine substitutions fail to facilitate the formation of invaginated membrane structures, suggesting that LETM1 plays a fundamental role in the organization of mitochondrial membrane morphology.

[1] Department of Molecular Biology, Graduate School of Medical Science, Kyushu University, Fukuoka 812-8582, Japan. [2] Department of Life Science, Rikkyo University, Tokyo 171-8501, Japan. [3] Department of Material and Biological Chemistry, Faculty of Science, Yamagata University, Yamagata 990-8560, Japan. [4] Department of Membrane Biochemistry, Okayama University Graduate School of Medicine, Dentistry and Pharmaceutical Sciences, Okayama 700-8530, Japan. [5] Laboratory for Infectious Immunity, RIKEN Research Center for Allergy and Immunology, Kanagawa 230-0045, Japan. [6] Laboratory of Biomolecular Science, Faculty of Pharmaceutical Sciences, Hokkaido University, Sapporo 060-0812, Japan. [7] Faculty of Life Sciences, Kyoto Sangyo University, Kyoto 603-8555, Japan. [8] These authors contributed equally: Seiko Nakamura, Aiko Matsui, Shiori Akabane. ✉email: toka@rikkyo.ac.jp

The human *LETM1* (*L*eucine-zipper and *E*F-hand-containing *T*rans*M*embrane protein *1*) gene was originally identified as one of the genes in the chromosomal region deleted in patients with Wolf–Hirschhörn syndrome[1], whose clinical features include mental retardation, a characteristic facial appearance, delayed growth and development, and seizures[2]. *LETM1* codes for a mitochondrial inner membrane protein and is evolutionarily conserved among eukaryotes[3–5]. Gene deletion of *MDM38*, a yeast LETM1 homolog, causes aberrant mitochondrial morphology and growth defects on non-fermentable carbon sources[6,7]. Knockdown of *LETM1* induces mitochondrial swelling in worms, flies, and protists[3,8,9]. Homozygous deletion of *LETM1* results in embryonic lethality in mice[10]. In addition to *LETM1*, paralog genes, called *LETM2*, are found in yeast, plants, and mammals[4,11,12]. Paralogs in yeast and plants have partially redundant roles with *LETM1*, whereas mammalian *LETM2* is expressed exclusively in the testis and sperm, suggesting that it has a specific role in spermatogenesis[4].

The *LETM1* gene possesses four typical sequence motifs (mitochondria-targeting sequence [MTS], transmembrane, leucine zipper, and EF hand), the first two of which manifest because of a mitochondrial membrane protein. Although the leucine-zipper and EF-hand domains are well-known motifs, their functional significance is not well investigated. In addition, a long segment including the transmembrane that is deposited as a domain architecture (pfam07766) in the Pfam database is highly conserved not only among LETM1 orthologs but also in LETM2 proteins[7,11,12].

Extensive studies of human LETM1 and its orthologs indicate potential roles in a variety of mitochondrial events. The yeast Mdm38 protein has a vital role in potassium ion homeostasis[13] and is also reported to act as a receptor for membrane-bound mitochondrial ribosomes[11]. In mammals, LETM1 exhibits calcium ion transport activity in vitro and in cell[10,14–17]. Interestingly, monovalent cation homeostasis in cell also requires LETM1[18]. In contrast to the widespread physiologic roles of LETM1 in mitochondrial events, changes in mitochondrial morphology, i.e., mitochondrial swelling and aberrant cristae structures, are common features resulting from the downregulation of LETM1 orthologs[3–5,7–10].

In the present study, we demonstrate that a LETM1 recombinant protein is sufficient to facilitate the formation of invaginated membrane structures in giant artificial liposomes in vitro. Furthermore, we identified the conserved segment of LETM1 that is indispensable for complementing the growth defects of yeast *mdm38* mutants. Two kinds of LETM1 mutant proteins in which four amino acid residues were altered lost the ability to change the shapes of proteoliposomes, suggesting that LETM1 is involved in forming the mitochondrial membrane morphology independent of mitochondrial ion homeostasis.

## Results

### The C-terminal region of LETM1 is oriented toward the matrix.

LETM1 is a mitochondrial inner membrane protein with a single transmembrane domain, and protease protection analysis revealed that the large C-terminal domain of LETM1 is exposed to the matrix side[3–5]. To verify the membrane topology of LETM1, we performed immunofluorescence analysis using semi-permeabilized cells and a monoclonal antibody specific to the C-terminal domain of LETM1 (Fig. 1a). On the basis of the observed differences in the tolerance of membranes to the detergent, organellar membranes (plasma membrane and mitochondrial outer and inner membranes) are stepwise permeabilized by treatment with different concentrations of digitonin[19]. Antibodies to the outer membrane protein Tom20 stained

mitochondria at any concentration of digitonin, whereas antibodies to the intermembrane space protein, cytochrome c, and the matrix protein, mtHSP70, recognized their antigens only when treated with higher concentrations of digitonin (>0.4 mg/ml for cytochrome c, 2.0 mg/ml for mtHSP70; Fig. 1c). Antibodies specific for the C-terminal domain of LETM1 had access to LETM1 only at 2.0 mg/ml digitonin (Fig. 1c), indicating that the C-terminal domain of LETM1 is present inside the mitochondrial inner membrane. Thus we confirmed that the C-terminal domain of LETM1 is oriented toward the matrix side (Fig. 1b).

### Increased LETM1 expression induces aberrant cristae structures.

Knockdown of LETM1 induces mitochondrial swelling and the disappearance of cristae structures[3,4]. To examine the effect of increased LETM1 expression on mitochondrial morphology without fixing the cells, we introduced a plasmid expressing the monomeric green fluorescent protein-fused LETM1 (LETM1-GFP) into HeLa cells and vitally stained the cells with a mitochondria-specific fluorescent dye Mitotracker. Live images of transfected cells showed that ectopic expression of LETM1-GFP induced mitochondrial fragmentation (Fig. 1d), consistent with findings from a previous study[4]. The degree of mitochondrial fragmentation appeared to correlate with the GFP intensity, i.e., the LETM1 expression level (Fig. 1d). Similar results were obtained when non-tagged LETM1 protein was expressed ectopically (Supplementary Fig. 1a). Overexpression of exogenous mitochondrial membrane proteins often leads to changes in mitochondrial morphology[20]. Expression of LETM1-GFP was comparable to that of endogenous LETM1 (Supplementary Fig. 1b), however, suggesting that the mitochondrial fragmentation was not due to excess amounts of the exogenous protein. To investigate the membrane structures of mitochondria of cells expressing LETM1-3HA, we performed immunoelectron microscopy by rapid freezing and freeze-substitution method, which is suitable for morphological preservation rather than chemical fixation[21]. Expression of LETM1-3HA was 55% lower than that of endogenous LETM1 (Supplementary Fig. 1c) and its product was not rapidly degraded (Supplementary Fig. 1d). Gold particles were clearly found on cristae membranes of LETM1-3HA-expressing cells (Fig. 1e, arrowhead) compared with untransfected cells. Most of gold particles in mitochondria were observed on the interior of mitochondria (Fig. 1f and Supplementary Data 1), suggesting that LETM1 was an inner membrane protein localized mainly in the cristae membranes. Interestingly, certain cells ectopically expressing LETM1 had smaller mitochondria and the matrix, rather than the inner membranes, was electron dense, i.e., the matrix was darker and the inner membranes were lighter (Fig. 1e, arrow). Such condensed mitochondria were frequently observed in LETM1-3HA-expressing cells compared with untransfected cells (Fig. 1i and Supplementary Data 1), suggesting that upregulation of LETM1 induced mitochondrial shrinkage. Mild cell fixation is used for immunoelectron microscopy to preserve antigens, allowing for lower-contrast images of the membranes. Therefore, we used electron microscopy to examine cells transfected with the LETM1 expression plasmid by standard fixation without immunostaining. Electron-dense mitochondria were observed in some cells and exhibited mesh-patterned inner membrane structures (Fig. 1g). Each area enclosed by the meshed inner membranes showed distinct electron densities (Fig. 1g, arrowheads), suggesting that the matrix was compartmentalized by the invaginated inner membranes. The numbers of cristae junctions remained unchanged in the LETM1-3HA-expressing cells (Fig. 1h and Supplementary Data 1), suggesting that formation of the meshed inner membranes was not due to changes in the number of the invaginated membranes. A high-magnification

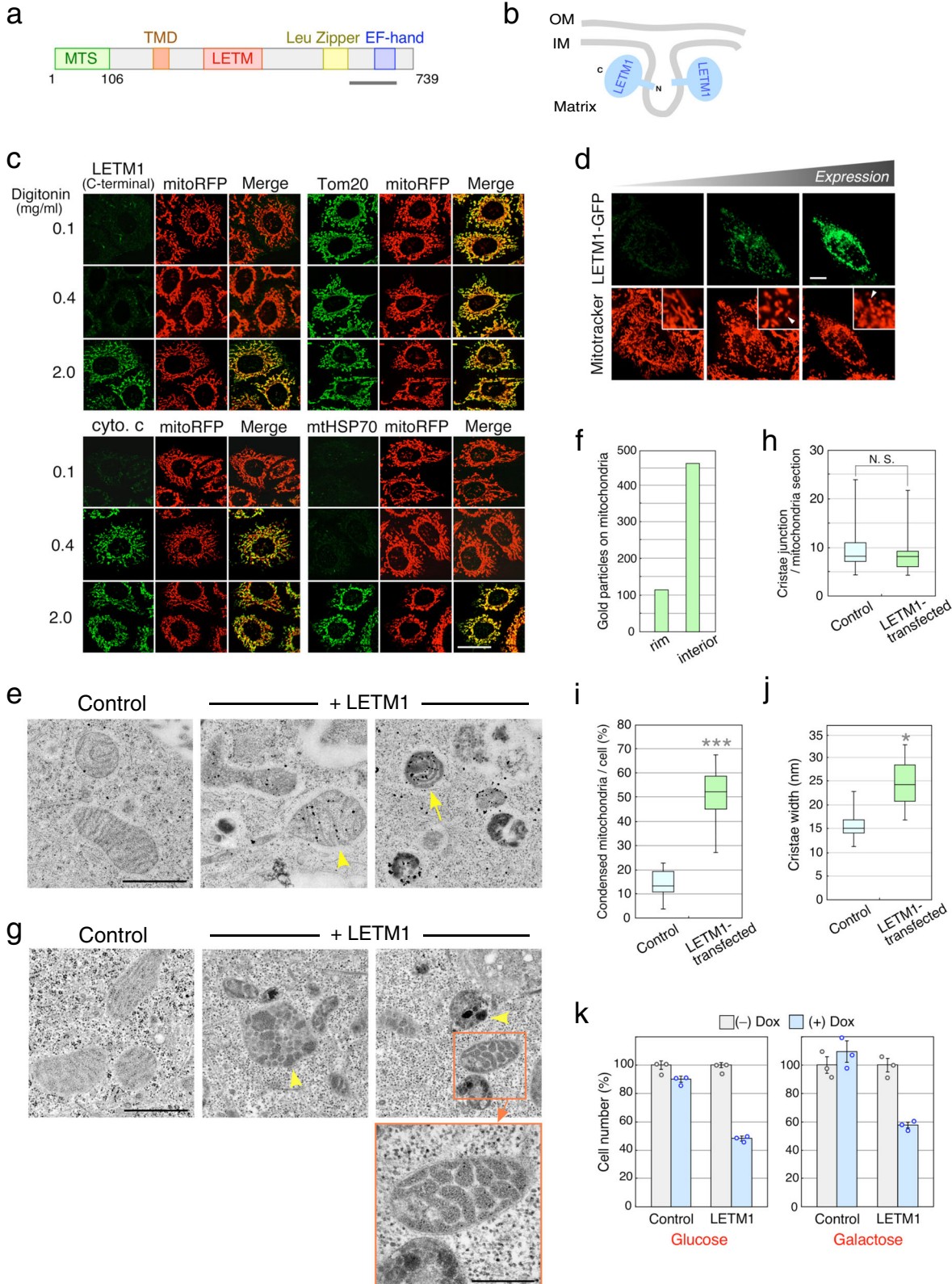

image showed an increase in the width of the meshed membranes (Fig. 1g), compared with cristae membranes in untransfected cells. In LETM1-3HA-expressing cells, the mean cristae width increased approximately 1.7-fold (Fig. 1j and Supplementary Data 1). Taken together with the absence of an influence on the number of cristae junctions, these results suggest that the meshed membrane structures are formed by the adhesion of two adjacent invaginated

membranes. Formation of OPA1 oligomers is linked to changes in cristae structures and OPA1 oligomerization is stimulated in mitochondria with narrow cristae structures[22]. We examined the effects of LETM1 ectopic expression on the formation of OPA1 oligomers using a membrane-permeable cross-linker. As expected, LETM1 knockdown induced a reduction in OPA1 oligomer formation (Supplementary Fig. 1f), because LETM1 downregulation

**Fig. 1 Increased LETM1 expression induces mitochondrial shrinking and aberrant cristae structures. a** Scheme of LETM1. Green, orange, red, yellow, and purple boxes represent the mitochondria-targeting sequence (MTS), transmembrane domain (TMD), LETM domain, leucine zipper, and EF-hand motif, respectively. Bar indicates the position of an antigen for a monoclonal antibody specific to the LETM1 C-terminal domain. **b** Topology of LETM1. The LETM1 C-terminal domain is exposed to the matrix side. **c** Cells stably expressing mito-RFP were fixed, permeabilized with digitonin at the indicated concentrations, and immunostained with antibodies to the LETM1 C-terminal domain, Tom20, cytochrome c (cyto. c), and mtHSP70. Scale bar, 20 μm. **d** LETM1-GFP-transfected cells were stained with Mitotracker. Live images of untransfected and transfected cells were obtained by confocal microscopy. Arrowheads indicate fragmented mitochondria. Scale bar, 10 μm. **e** Control cells and LETM1-3HA-transfected cells were fixed and subjected to immunoelectron microscopy using anti-HA antibody and gold particle-conjugated antibody. Arrowhead and arrow represent mitochondria with gold particles on the cristae membranes and a shrinking mitochondrion, respectively. Scale bar, 1 μm. **f** Distribution of gold particles of LETM1-3HA-transfected cells was scored. Thirty individual mitochondria were analyzed. **g** Control cells and LETM1-3HA-tranfected cells were fixed, and thin sections were visualized by electron microscopy. High-magnification image is shown. Arrowheads indicate mitochondria with aberrant cristae structures. Scale bars, 1 μm and 500 nm (high magnification). **h–j** Cristae junctions per mitochondrion (**h**), electron-dense mitochondria per cell (**i**), and cristae width (**j**) of the sections described in **g** were quantified. Center lines, boxes, and whiskers represent the median, the interquartile range, and the full extent of the data of three independent experiments, respectively; >20 individual mitochondria were counted. Statistical significance was calculated using Student's t test. *$p < 0.0005$, ***$p < 0.000001$, N.S. not significant ($p > 0.05$). **k** Cells stably expressing chloramphenicol acetyltransferase (control) or LETM1-3HA were treated with doxycycline (Dox) and cultured in medium containing either glucose or galactose. Data represent the mean ± SEM of three independent experiments.

causes mitochondrial swelling and disruption of cristae structures[4]. Ectopic expression of LETM1, however, did not affect OPA1 oligomerization (Supplementary Fig. 1e), supporting the idea that the increased cristae width in the meshed membranes was due to the adhesion of the invaginated membranes and not to dilated cristae structures. These findings suggest that increased LETM1 expression induces mitochondrial shrinkage with tight adhesions of the inner membranes.

**LETM1 upregulation causes cellular and mitochondrial damage.** Mitochondrial fragmentation influences membrane potential and cell growth[23]. To examine the effect of exogenous LETM1 expression on the membrane potential, cells transfected with LETM1-GFP were stained with the membrane potential-dependent fluorescent dye, tetramethylrhodamine methyl ester (TMRM). The LETM1-GFP-expressing cells were barely stained with TMRM compared with untransfected cells (Supplementary Fig. 2a), indicating that cells expressing exogenous LETM1 had low mitochondrial membrane potentials. Production of mitochondrial reactive oxygen species (ROS) is affected by changes in the membrane potential[24]. To test whether the ectopic expression of LETM1 induces ROS production, LETM1-GFP-transfected cells were stained with the fluorescent dye MitoSOX Red, a membrane-permeable indicator for monitoring mitochondrial ROS. Thirty-three percent of cells expressing LETM1-GFP were clearly stained with MitoSOX Red (Supplementary Fig. 2b), indicating that exogenous expression of LETM1 led to increased ROS production. Moreover, cells stably expressing LETM1-3HA under an inducible promoter exhibited poor cell growth in medium containing different carbon sources (Fig. 1k and Supplementary Data 1). Growth defects were observed even when both exogenous and endogenous LETM1 proteins were equally expressed (Supplementary Fig. 2c). These findings suggest that upregulation of LETM1 expression causes cellular and mitochondrial damage through changes in the cristae structures.

**The LETM domain is crucial for mitochondrial morphology.** The *LETM1* gene was originally discovered as a gene present in the deleted chromosomal region of patients with Wolf–Hirschhörn syndrome[1], and no pathologic missense mutations have been reported. In addition to the characteristic motifs of LETM1[1,4,7,25], the C-terminal region proximal to the transmembrane, which is a part of the domain architecture deposited in the Pfam database, was highly conserved among LETM1 and LETM2 orthologs in yeast, worms, flies, plants, and mammals (Figs. 1a and 2a). Therefore, we named this conserved region the LETM domain. To determine the domains necessary for the functions of human LETM1, three LETM1 mutants (ΔLETM domain, ΔEF-hand domain, and alanine substitution in the leucine-zipper domain) were examined by complementation of the growth defects caused by deleting the yeast *MDM38* gene. Expression of human wild-type LETM1 complemented the growth defects of Δmdm38 cells on a non-fermentable carbon source (Fig. 2b), as reported previously[7]. Two LETM1 mutants (ΔEF-hand domain and alanine substitution in leucine-zipper domain) ameliorated the Δmdm38 growth deficiency, but the ΔLETM domain mutant did not restore the growth of Δmdm38 cells (Fig. 2b). To identify the amino acid residues in the LETM domain responsible for complementing the Δmdm38 growth deficiency, we selected three sets of highly conserved amino acid triplets containing at least one charged residue (Fig. 2a, asterisks) and substituted alanine for all of the amino acid residues. The nonuple alanine-substituted mutants did not suppress the Δmdm38 growth defect, indicating that the three sets of amino acid triplets contained essential residues for LETM1 function. We divided the nonuple alanine mutations into single and triple alanine mutants and identified two missense mutants of human LETM1 (Fig. 2b, and Supplementary Fig. 3a, b). The LETM1-1 triple mutant (R382A/G383A/M384A) clearly lost the ability to complement the Δmdm38 growth deficiency, whereas the LETM1-2 single mutant (D359A) slightly supported the growth of Δmdm38 cells on minimal essential medium (SCG) at 30 °C (Fig. 2b) but not on rich medium (YPG) at 37 °C (Supplementary Fig. 3a).

Yeast Δmdm38 cells exhibit an aberrant mitochondrial morphology such as swelling and/or dilated filaments[6,7]. We assessed the effects of two LETM1 missense mutants on the mitochondrial morphology of Δmdm38 cells. Expression of wild-type LETM1 suppressed the morphologic defects of mitochondria in Δmdm38 cells as well as the expression of the yeast *MDM38* gene. Two LETM1 mutants, however, failed to restore the mitochondrial morphology in Δmdm38 cells (Fig. 2c, Supplementary Fig. 4a, and Supplementary Data 1). Furthermore, electron microscopy showed that the cristae structures were clearly restored by the expression of wild-type LETM1 but not by the expression of the two missense mutants (Fig. 2d and Supplementary Fig. 4b, c). Thus the LETM1-1 and LETM1-2 missense mutants lost the LETM1 functions in mitochondrial morphology.

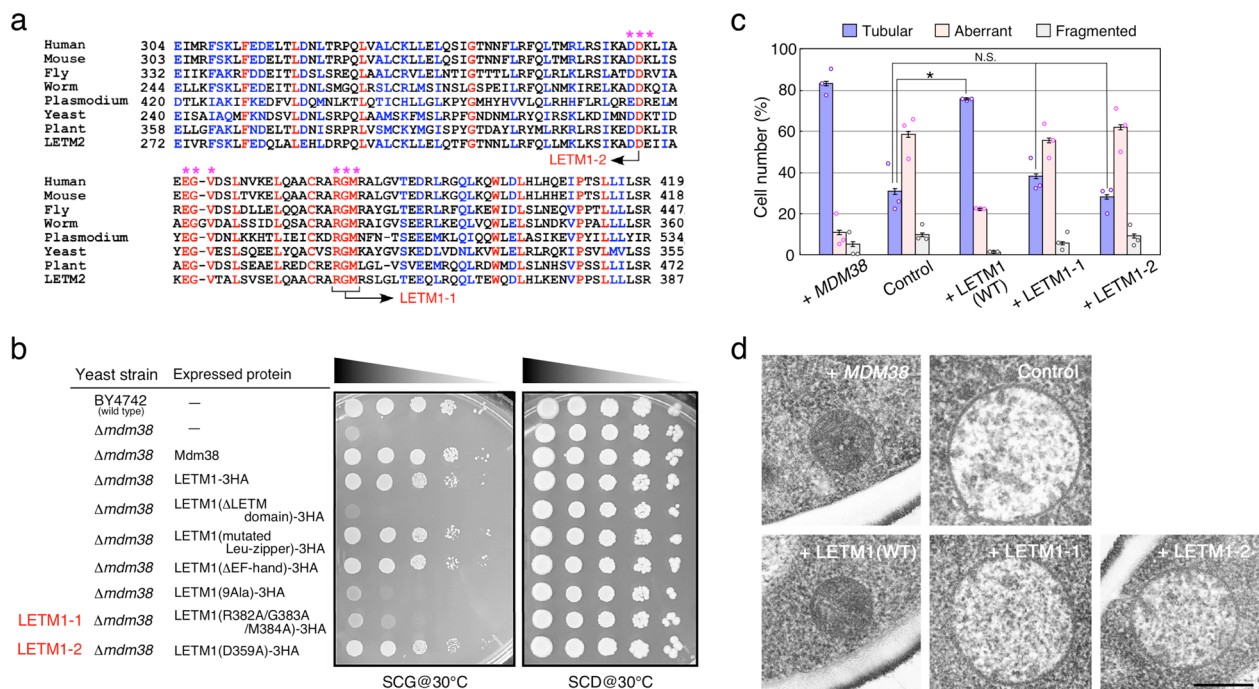

**Fig. 2 Identification of amino acid residues in the LETM domain that are responsible for growth complementation and morphologic defects in yeast *mdm38* mutants. a** Multiple alignments of amino acid sequences in the LETM domain among LETM1 orthologs. Red and blue characters indicate identical and conserved amino acid residues, respectively. Asterisks represent amino acid residues that were substituted with alanine in the LETM1 mutants. **b** Growth phenotypes of yeast *mdm38* mutants. Serial dilutions of wild-type (BY4742) and *mdm38* null mutant (*Δ mdm38*) cells expressing the indicated proteins were plated at 30 °C onto minimal medium containing either glycerol (SCG) or dextrose (SCD). **c** Yeast *mdm38* mutant cells with control plasmid and the plasmid containing the indicated genes were transformed with a mitochondria-targeted GFP and then cultured at 30 °C onto SCD medium. Mitochondrial morphology of >50 individual cells was scored using live fluorescent images. Data represent the mean ± SEM of three independent experiments. Statistical significance was calculated using Student's *t* test. *$p < 0.005$; N.S. not significant ($p > 0.05$). **d** *mdm38* mutant cells with control plasmid and the plasmid containing the indicated genes were cultured at 30 °C onto SCD medium and mitochondrial membrane structures were analyzed by electron microscopy. Scale bar, 400 nm.

**The defects of LETM1 mutants in the complex assembly**. To evaluate the effect of the LETM1 missense mutants on mitochondrial morphology in mammalian cells, HeLa cells were transfected with vector and expression plasmids carrying wild-type or mutant LETM1-GFP and stained with Mitotracker. As expected, wild-type LETM1-GFP clearly induced mitochondrial fragmentation (Fig. 3a, arrowhead). Neither LETM1 missense mutant significantly affected the mitochondrial morphology (Fig. 3a, b, and Supplementary Data 1), however, despite the comparable expression level of the mutant proteins with that of the wild-type protein (Supplementary Fig. 1b), confirming that the LETM1-1 and LETM1-2 missense mutants barely influence mitochondrial morphology in mammalian cells.

LETM1 forms oligomers with apparent molecular weights ranging from 250 to 500 kDa on native gels[3–5]. The size heterogeneity of the oligomers appeared to depend on the electrophoretic conditions including detergents and Coomassie dyes (Supplementary Fig. 5a). To examine the oligomerization of LETM1 missense mutants, mitochondrial fractions were prepared from cells transfected with the expression plasmid of either wild-type or mutant LETM1-3HA and analyzed by clear-native polyacrylamide gel electrophoresis (PAGE) and immunoblotting. The wild-type LETM1 migrated as a distinct band with an apparent molecular size of 250 kDa, whereas the LETM1-1 and LETM1-2 mutant proteins failed to migrate as only a single band and formed smeared bands with apparent molecular weights ranging from 250 to >1000 kDa (Fig. 3c and Supplementary Fig. 6), suggesting that LETM1 missense mutations repress the correct assembly of LETM1-containing protein complexes.

**LETM1 facilitates in vitro membrane invagination**. LETM1 downregulation induces mitochondrial swelling and the disappearance of cristae structures[3–5]. Increased LETM1 expression caused mitochondrial shrinkage and meshed cristae structures (Fig. 1e). The relationship between the LETM1 gene dosage and changes in the cristae structures led us to hypothesize that LETM1 acts directly on the formation of membrane structures. To test this possibility, we attempted in vitro reconstitution of giant proteoliposomes containing LETM1. His-tagged LETM1 recombinant proteins lacking MTS (His-LETM1 in Fig. 4a) were expressed in silkworms, which efficiently express mitochondrial inner membrane proteins[26]. Because endogenous LETM1 was solubilized with detergents in the presence of >300 mM NaCl (Supplementary Fig. 5b), wild-type and mutant proteins expressed in silkworms were solubilized under the same conditions and purified by affinity column chromatography (Fig. 4b). Clear-native PAGE and Coomassie Brilliant Blue staining showed that wild-type His-LETM1 proteins migrated as regularly spaced, ladder bands with molecular sizes ranging from 500 to 1100 kDa (Fig. 4b, asterisks), indicating that LETM1 can oligomerize and homo-oligomers can further interact to form heterogeneous-sized complexes. Both the His-LETM1-1 and His-LETM1-2 mutant proteins formed smeared bands with larger molecular sizes in addition to a single band with an apparent molecular size of 700 kDa (Fig. 4b, bracket), suggesting that purified LETM1 mutant proteins retain the ability to form oligomers, but the LETM1 mutant oligomers may lose the ability to interact correctly with each other.

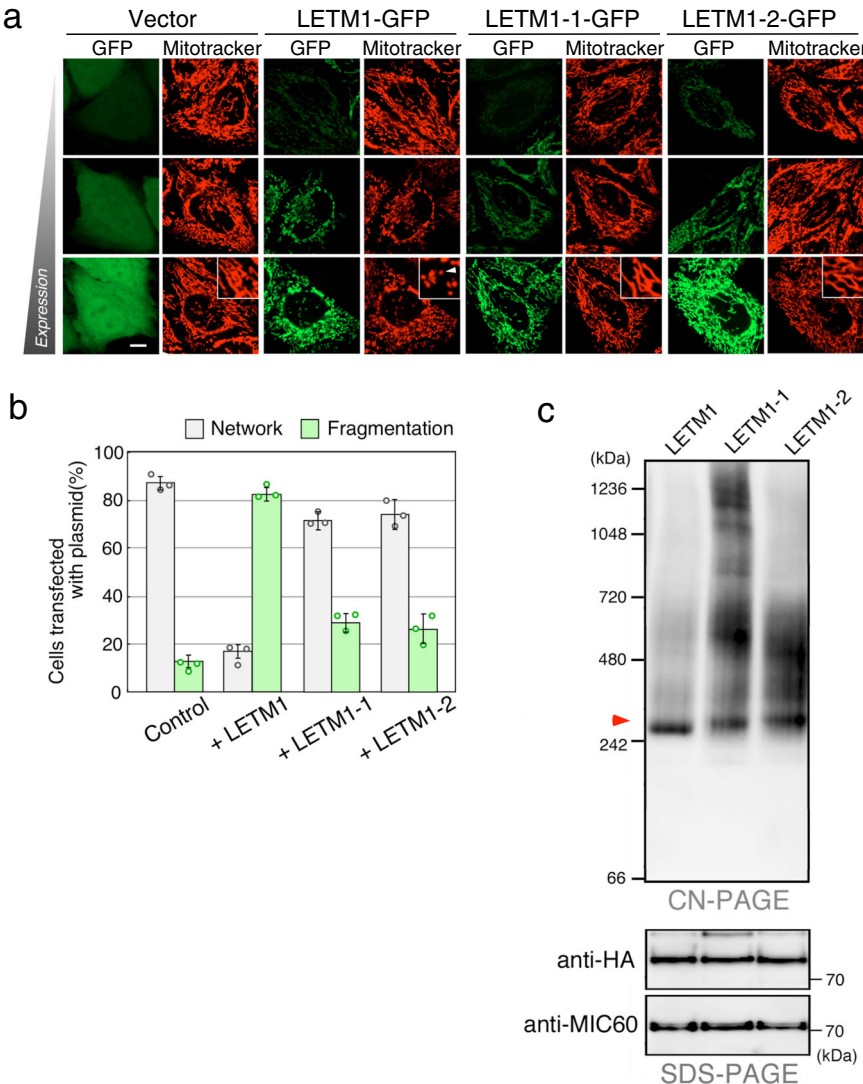

**Fig. 3 LETM1 mutant proteins failed in the correct assembly of LETM1 complexes. a** Cells were transfected with vector and expression plasmids carrying wild-type or mutant LETM1-GFP and stained with Mitotracker. Live fluorescent images were obtained by confocal microscopy. High-magnification images are superimposed. Arrowhead indicates fragmented mitochondria. Scale bar, 10 μm. **b** Mitochondrial morphology in the cells transfected with plasmids carrying wild-type or mutant LETM1-3HA was analyzed. Data represent the mean ± SEM of three independent experiments; >100 individual cells were counted. **c** Digitonin-solubilized mitochondrial fractions were prepared from wild-type and mutant LETM1-3HA-expressing cells and analyzed by Clear-native PAGE or SDS-PAGE, followed by immunoblotting using antibody to HA tag or MIC60. Red arrowhead indicates a 250-kDa protein complex containing LETM1.

To prepare giant proteoliposomes, purified His-LETM1 proteins were added to detergent-solubilized mixtures of the phospholipids that mainly comprise the mitochondrial inner membranes: phosphatidylcholine, phosphatidylethanolamine, phosphatidylinositol, and cardiolipin[27]. After 20-fold dilution with a detergent-free buffer, the resulting proteoliposomes were immediately fixed and analyzed by electron microscopy, because giant proteoliposomes with diameters >500 nm were quite unstable and rapidly disappeared within several minutes. In the proteoliposomes containing wild-type His-LETM1 proteins, a part of the liposome membrane was folded inward, and such inward membrane structures were frequently observed (Fig. 4c, arrowheads). Three-dimensional analysis of serial sections indicated that the folded membrane was an invaginated structure and not just a large curvature of the proteoliposome (Fig. 4d, Supplementary Fig. 5c, and Supplementary Movie 1). When wild-type LEMT1 was added, 26% of the proteoliposomes with a diameter >500 nm contained a membrane invagination (Fig. 4e

and Supplementary Data 1). The His-LETM1-1 mutant protein, however, did not form membrane invaginations in proteolipo-somes (Fig. 4c, e, and Supplementary Data 1). When His-LETM1-2 was added, only 7.3% of the proteoliposomes had invaginated membranes (Fig. 4e and Supplementary Data 1), consistent with the complementation assay of growth defects in yeast Δmdm38 cells (Fig. 2b). These findings strongly indicate that LETM1 is sufficient to induce membrane invagination in vitro. These findings, together with the fact that LETM1 missense mutants failed to form homo-oligomer ladder complexes, suggest that the correct association of LETM1 oligomers facilitates the formation of in vitro membrane invaginations.

## Discussion

In addition to the MTS and transmembrane domain, LETM1 has two well-known motifs (leucine zipper and EF hand) and an uncharacterized domain (LETM domain) in the C-terminal

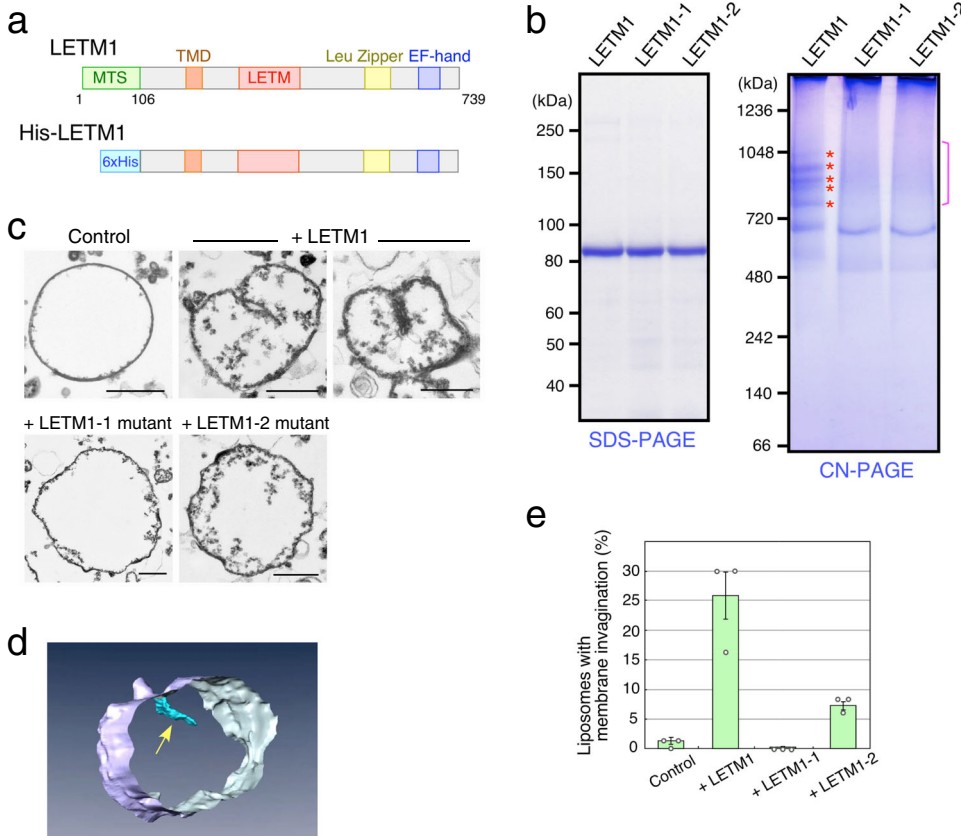

**Fig. 4 LETM1 mediates the formation of in vitro membrane invagination. a** Scheme of His-tagged LETM1 lacking MTS. **b** His-LETM1 recombinant proteins expressed in silkworms were purified and analyzed by Clear-native PAGE or SDS-PAGE, followed by Coomassie Brilliant Blue staining. **c** Proteoliposomes containing wild-type or mutant His-LETM1 purified proteins were produced and analyzed by electron microscopy. Scale bars, 500 nm. **d** Three-dimensional images were reconstituted by serial sections of a proteoliposome containing wild-type His-LETM1. Arrow indicates membrane invagination. **e** Proteoliposomes with membrane invagination as described in **c** were scored. Data represent the mean ± SEM of three preparations; 50 individual proteoliposomes >500 nm in diameter were counted.

region that is exposed to the matrix side. The LETM domain is a long highly conserved segment among LETM1 and LETM2 orthologs in eukaryotes[4]. Complementation assays using yeast cells demonstrated that the LETM domain, but not the leucine-zipper and EF-hand domains, was required to suppress the growth defect of the yeast *mdm38* gene-deletion mutant, indicating that the well-known characteristic motifs of LETM1 are not essential for LETM1 functions in yeast cell growth on a non-fermentable carbon source. We identified two types of LETM1 missense alleles in the LETM domain (LETM1-1 and LETM1-2) that lost the ability to complement the growth defect of the *mdm38* mutant. Multiple alignment of the protein sequences revealed that the four amino acid residues (D359, R382, G383, and M384) mutated in the missense alleles were identical among the LETM1 and LETM2 orthologs, suggesting that they are indispensable residues for the common functions of LETM1 and LETM2. According to the crystal structure of the C-terminal domain of the yeast monomeric Mdm38 protein[28], all four amino acid residues mutated in the LETM1 alleles are buried in the interior of the structure rather than located on the surface, suggesting that these residues contribute to maintain the domain conformation for the oligomer association rather than surface interactions. We cannot, however, rule out the possibility that the structure of LETM1 protein in the oligomeric complexes differs from the crystal structure of the yeast monomeric Mdm38 protein[28].

Exogenous expression of LETM1 induced mitochondrial fragmentation. Mitochondrial fission is governed by cytoplasmic GTPase Drp1[29]. Drp1 knockdown stimulates elongation of mitochondrial filaments[30,31]. In contrast, LETM1 knockdown causes large dot-like structures of mitochondria[3,4,6]. Simultaneous knockdown of Drp1 and LETM1 results in an additive phenocopy, i.e., both elongation and swelling of mitochondrial filaments[4,5], suggesting that LETM1 functions in mitochondrial morphology separately from the Drp1-mediated membrane fission pathway. Furthermore, LETM1 knockdown induces a reduction in mitochondrial membrane potential[4] and exogenous expression of LETM1 caused mitochondrial damage, including ROS generation and the low membrane potential, suggesting that changes in LETM1 expression may induce mitochondrial stress through morphological alterations of mitochondrial inner membranes. Therefore, mitochondrial fragmentation caused by LETM1 upregulation may be indirectly stimulated by mitochondrial stress.

Endogenous LETM1 protein oligomerizes with apparent molecular sizes of 250–500 kDa[3–5]. Purified His-LETM1 recombinant proteins tended to migrate ladder-like bands on clear-native PAGE gels in vitro and the complexes of His-LETM1 proteins were larger than those of endogenous LETM1 proteins observed in cell, suggesting that hyper-assembly of LETM1 complexes in mitochondrial inner membranes is enzymatically and spatially repressed by other proteins and/or lipids in cell. Indeed, a mitochondrial inner membrane AAA-ATPase, BCS1L,

regulates the assembly of LETM1 complexes[4]. Both LETM1 mutant proteins prepared from HeLa cells formed large and various sizes of complexes (250–1000 kDa), suggesting that large complexes of LETM1 mutant proteins may be insensitive to disassembly mediated by regulatory proteins.

On the basis of clinical studies of patients with Wolf–Hirschhörn syndrome, gene deletion of LETM1 in model organisms, including yeast, worms, mice, and plants, was strategically applied to explore the functions described in previous studies[3,7,10,12]. Studies using null mutations provided insights into the physiologic relevance of the LETM1 gene but provided no new information on the functions of the conserved domains. Our findings revealed that the four amino acid residues in the LETM domain identified by the complementation assay have crucial roles in the following: (1) maintaining mitochondrial morphology, (2) correct assembly of the LETM1 complexes, and (3) in vitro formation of invaginated membranes, which suggests that the LETM domain of LETM1 arranges the shapes of mitochondrial inner membranes via the formation of oligomeric LETM1 complexes. LETM1 protein is also involved in calcium ion transport activity in vitro and in cell[10,14–17]. In addition, LETM1 contributes to regulating mitochondrial calcium ion fluxes via monovalent cation homeostasis[18,32]. Therefore, changes in mitochondrial morphology in the LETM1-deficient cells could arise from the possible imbalance of mitochondrial ion homeostasis. Meanwhile, we demonstrated that an increase in LETM1 expression caused an aberrant cristae morphology and LETM1 recombinant proteins induced in vitro membrane invagination of artificial liposomes, suggesting that the role of LETM1 in mitochondrial morphology could be separable from that of LETM1 in ion homeostasis. In other words, LETM1 may play dual roles in organizing mitochondrial inner membranes and maintaining ion homeostasis. Transporter proteins with additional functions in regulation of membrane morphology are not unprecedented. For example, ATP synthase, a protein complex that functions as a proton transporter for the cellular ATP synthesis, directly contributes to the shape of lamellar cristae[33,34]. Furthermore, dimerization of ATP synthase facilitates the formation of the highly curved edges of cristae structures[35–37]. Likewise, LETM1 may, in addition to the functions in mitochondrial ion transport, manipulate the shapes of mitochondrial inner membranes through its oligomeric complex formation.

Several lines of evidence indicate that a few factors including ATP synthase, OPA1, and MICOS complex directly and indirectly contribute to the formation and maintenance of cristae membrane structures[38–41]. Dimeric rows of ATP synthase are observed along cristae edges and contribute to the formation of the membrane curvature at the edges[36,37]. The OPA1 and MICOS complex are localized at the junctions between the cristae and inner boundary membranes, and loss of either the OPA1 or MICOS complex results in the disappearance of the cristae junctions[38,40]. LETM1 knockdown causes the disappearance of the cristae structures and mitochondrial swelling[3–5]. Furthermore, upregulation of LETM1 led to formation of the meshed cristae membranes and mitochondrial shrinkage, suggesting that LETM1 facilitates adhesion of neighboring inner membranes. In addition, LETM1 recombinant proteins formed invaginated membranes in vitro, indicating that LETM1 proteins can facilitate the adherence of nearest parts of membranes within single artificial liposomes. Thus LETM1 may mediate the formation of appropriate adhesion of mitochondrial inner membranes to maintain cristae structures. Regulatory proteins including AAA-ATPase BCS1L may control the assembly of excess LETM1 complexes to ensure the flexibility of cristae membrane structures. It is possible, however, that LETM1 helps to form the curved membranes at the junctions and edges of the cristae

structures. Further investigation will provide insight into the molecular mechanisms and physiologic roles of LETM1 in the membrane biogenesis of cristae structures.

## Methods

**Human cell line and DNA transfection**. HeLa cells and HeLa cells expressing stably mitochondria-targeted red fluorescent protein[42] were maintained at 37 °C in Dulbecco's modified Eagle's medium (DMEM) supplemented with 10% fetal bovine serum and 4.5 mg/ml glucose. HEK cell line expressing stably LETM1-3HA under an inducible promoter was cultured in the DMEM medium. For induction of LETM1-3HA, 1 μg/ml doxycycline was added to the medium. The plasmids used in this study are summarized in Table 1. DNA transfection was performed using FuGENE HD (Promega) according to the manufacturer's instructions. For vital staining, cells were stained with either TMRM (20 nM) or Mitotracker Red (100 nM) in the DMEM medium. Live fluorescent images of mitochondrial morphology were obtained using an optical sectioning microscope, Viva Tome (Carl Zeiss Inc.), or a confocal laser-scanning microscope LSM510 (Carl Zeiss Inc.).

**Growth condition of yeast strains**. Yeast strains used in this study are summarized in Table 2. Yeast cells were grown at the indicated temperatures in YPD (1% yeast extract, 2% polypeptone, and 2% glucose), YPG (1% yeast extract, 2% polypeptone, and 3% glycerol), SCD (0.67% yeast nitrogen base, 0.5% casamino acids, 2% glucose), or SCG (0.67% yeast nitrogen base, 0.5% casamino acids, 3% glycerol).

**Construction of plasmids**. To construct the expression plasmids for LETM1-GFP, a DNA fragment containing LETM1 was digested from pEF1-LETM1-3HA[4] and inserted into pAcGFP-N1 (Clontech). LETM1-1 and LETM1-2 mutants were constructed by PCR using the primers (LETM1-1, 5′-ggcagcagcagcacgggccctgggcg tcacggaa-3′ and 5′-gcccgtgctgctgctgcccgacacgctgcctgcag-3′; LETM1-2, 5′-ggcagacg ctaagctgattgctgaggaaggggtg-3′ and 5′-aatcagcttagcgtctgcctttatggagcgcagc-3′) and cloned into pEF1-3HA and pAcGFP-N1.

For the expression of human LETM1 in yeast cells, the ADH1 promoter and CYC1 terminator of pMID-2[43] were subcloned into a yeast multicopy vector pRS426 to construct pADH1. A DNA fragment of LETM1-3HA from pEF1-LETM1-3HA was inserted into pADH1 to construct pADH1-LETM1-3HA. A 2.5-kb DNA fragment containing the yeast MDM38 gene and its promoter region was amplified by PCR from a yeast genomic DNA using the primers (5′-acttgtgcagagccttgtatcc-3′ and 5′-aaaaggtacccagctttactgcgtgcattac-3′) and cloned into a single-copy vector pRS316 to construct pRS316-MDM38.

**Immunofluorescence microscopy of semi-intact cells**. Immunofluorescence microscopy using semi-intact cells was carried out essentially as described previously[19]. Cells were fixed with 4% paraformaldehyde and then permeabilized for 5 min with 0.1, 0.4, or 2.0 mg/ml digitonin in buffer A [20 mM HEPES–KOH (pH 7.5), 250 mM sucrose, 2.5 mM magnesium acetate, 25 mM KCl, 1% bovine serum albumin, and 1 μM taxol]. The permeabilized cells were immunostained with the following primary antibodies: LETM1 (Abnova, clone 6F7), Tom20 (Santa Cruz Biotechnology, clone F-10), mtHSP70 (Stressgen, clone BB70), and cytochrome c (BD Pharmingen, clone 6H2.B4). Images were acquired using a confocal laser-scanning microscope LSM510 (Carl Zeiss Inc.).

**Clear-native PAGE**. Clear-native PAGE was performed essentially as described previously[44]. Mitochondria fractions (1 mg/ml) prepared from HeLa cells were solubilized at 4 °C for 30 min with 1% digitonin in buffer B [20 mM Bis-Tris (pH 7.0), 0.1 M NaCl, and 10% glycerol] and centrifuged at 100,000 × g for 15 min. The solubilized proteins were subjected to electrophoresis on 4–16% gradient gels in the running buffer [50 mM Tricine (pH 7.0) and 7.5 mM imidazole] containing 0.05% sodium deoxycholate (Dojindo Inc.) and 0.004% dodecyl-β-D-maltoside (Dojindo Inc.). After electrophoresis, the gels were incubated at 60 °C for 20 min with denaturing buffer [20 mM Tris-HCl (pH 6.8), 1% sodium dodecyl sulfate, and 0.1 M β-mercaptoethanol], and the proteins were then transferred to polyvinylidene difluoride membranes and subjected to immunoblotting analysis.

**Immunoblotting**. Immunoblotting was performed essentially as described previously[20]. After PAGE, the proteins were transferred to polyvinylidene difluoride membranes (MilliporeSigma) and immunoblotted with the indicated primary antibodies using the Immun-Star AP Chemiluminescence Kit (Bio-Rad) according to the manufacturer's instructions. The following primary antibodies were used: LETM1[4], HA tag (Roche, clone 3F10; Wako Chemicals, clone 4B2), and MIC60[45].

**Chemical cross-linking**. Cross-linking was carried out essentially as described previously[22]. Cells were harvested, washed with phosphate-buffered saline (PBS), and incubated with 1 mM bismaleimidohexane (Thermo Fisher Scientific) at 37 °C for 20 min. After termination by addition of 0.1% β-mercaptoethanol, cells were washed with PBS containing 0.1% β-mercaptoethanol twice and solubilized by the

**Table 1 Plasmids used in this study.**

| Plasmid | Description | Source |
|---|---|---|
| pNucScrIII-LETM1 | For simultaneous expression of LETM1 and nuclear-targeted GFP | 4 |
| pEF1-LETM1-3HA | For expression of LETM1-3HA | 4 |
| pEF1-LETM1 (R382A/G383A/M384A)-3HA | For expression of LETM1(R382A/G383A/M384A)-3HA | This study |
| pEF1-LETM1(D359A)-3HA | For expression of LETM1(D359A)-3HA | This study |
| pAcGFP-N1-LETM1 | For expression of LETM1 of which GFP was fused to the C-terminus | This study |
| pAcGFP-N1-LETM1(D359A) | For expression of LETM1(D359) of which GFP was fused to the C-terminus | This study |
| pAcGFP-N1-LETM1(R382A/G383A/M384A) | For expression of LETM1(R382A/G383A/M384A) of which GFP was fused to the C-terminus | This study |
| pRS316-MDM38 | For expression of Mdm38 in yeast cells, MDM38 gene is driven by its own promoter on a single copy plasmid | This study |
| pADH1-LETM1-3HA | pRS426-ADH1p-LETM1-3HA; for expression of LETM1-3HA in yeast cells, human LETM1-3HA is driven by the yeast ADH1 promoter on a multicopy plasmid | This study |
| pADH1-LETM1(9Ala[a])-3HA | pRS426-ADH1p- LETM1(9Ala)-3HA; for expression of LETM1(9Ala)-3HA in yeast cells, human LETM1(9Ala)-3HA is driven by yeast ADH1 promoter on a multicopy plasmid | This study |
| pADH1-LETM1(ΔLETM domain[b])-3HA | pRS426-ADH1p-LETM1(ΔLETM domain)-3HA; for expression of LETM1(ΔLETM domain)-3HA in yeast cells, human LETM1(ΔLETM domain)-3HA is driven by yeast ADH1 promoter on a multicopy plasmid | This study |
| pADH1-LETM1(ΔEF-hand[c])-3HA | pRS426-ADH1p-LETM1(ΔEF-hand)-3HA; for expression of LETM1(ΔEF-hand)-3HA in yeast cells, human LETM1(ΔEF-hand)-3HA is driven by yeast ADH1 promoter on a multicopy plasmid | This study |
| pADH1-LETM1(mutated Leu-zipper[d])-3HA | pRS426-ADH1p-LETM1(mutated Leu-zipper)-3HA; for expression of LETM1(mutated Leu-zipper)-3HA in yeast cells, human LETM1(mutated Leu-zipper)-3HA is driven by yeast ADH1 promoter on a multicopy plasmid | This study |
| pADH1-LETM1 (D358A/D359A/K360A)-3HA | pRS426-ADH1p-LETM1(D358A/D359A/K360A)-3HA; for expression of LETM1(D358A/D359A/K360A)-3HA in yeast cells, human LETM1(D358A/D359A/K360A)-3HA is driven by yeast ADH1 promoter on a multicopy plasmid | This study |
| pADH1-LETM1 (E365A/G366A/V367A)-3HA | pRS426-ADH1p-LETM1(E365A/G366A/V367A)-3HA; for expression of LETM1(E365A/G366A/V367A)-3HA in yeast cells, human LETM1(E365A/G366A/V367A)-3HA is driven by yeast ADH1 promoter on a multicopy plasmid | This study |
| pADH1-LETM1 (R382A/G383A/M384A)-3HA | pRS426-ADH1p-LETM1(R382A/G383A/M384A)-3HA; for expression of LETM1(R382A/G383A/M384A)-3HA in yeast cells, human LETM1(R382A/G383A/M384A)-3HA is driven by yeast ADH1 promoter on a multicopy plasmid | This study |
| pADH1-LETM1(D359A)-3HA | pRS426-ADH1p-LETM1(D359A)-3HA; for expression of LETM1(D359A)-3HA in yeast cells, human LETM1(D359A)-3HA is driven by yeast ADH1 promoter on a multicopy plasmid | This study |
| pFastbac1-His-LETM1(107-739) | For expression of His-LETM1(107-739) recombinant protein in silkworm | This study |
| pFastbac1-His-LETM1(107-739, R382A/G383A/M384A) | For expression of His-LETM1(107-739, R382A/G383A/M384A) recombinant protein in silkworm | This study |
| pFastbac1-His-LETM1 (107-739, D359A) | For expression of His-LETM1(107-739, D359A) recombinant protein in silkworm | This study |

[a]9Ala represents a nonuple mutation (D358A/D359A/K360A/E365A/G366A/V367A/R382A/G383A/M384A).
[b]ΔLETM domain represents a deletion of the LETM domain (ΔK230-L466).
[c]ΔEF-hand represents a C-terminus-truncated mutation (ΔD676-S739).
[d]Mutated Leu-zipper represents a quadruple mutation (L548A/L555A/L562A/L569A).

**Table 2 Yeast strain used in this study.**

| Strain | Genotype | Source |
|---|---|---|
| BY4742 | MATα his3Δ1 leu2Δ0 lys2Δ0 ura3Δ0 | Open biosystems |
| Δmdm38 | MATα his3Δ1 leu2Δ0 lys2Δ0 ura3Δ0 mdm38::KanMX6 | Open biosystems |

lysis buffer. The cell lysates were subjected to electrophoresis onto NuPAGE Tris-acetate gradient gel (Thermo Fisher Scientific) and immunoblotting.

**In vitro reconstitution of membrane invagination**. To prepare artificial liposomes with the four phospholipids (phosphatidylcholine [PC], phosphatidylethanolamine [PE], phosphatidylinositol [PI], and cardiolipin [CL]) that mainly comprise the mitochondrial inner membranes, phospholipids with a lipid composition similar to that of the mitochondrial inner membrane (PC/PE/PI/CL = 45/28/17/10)[27] were mixed, dissolved in chloroform, and dried with nitrogen gas. Dry phospholipid films were dissolved in detergent solution (20 mM Tris-HCl [pH 7.4], 2% n-octyl-β-D-glucoside; Dojindo Inc.) and sonicated to disrupt multi-lamellar liposomes. The resulting solution containing small unilamellar liposomes was mixed with purified His-LETM1 protein and diluted 20 times with the

reconstitution buffer (20 mM Tris-HCl [pH 7.4]) 100 mM NaCl, 5 mg/ml lysozyme) to make giant liposomes. After adding a one-third volume of buffer E (20 mM Tris-HCl [pH 7.4], 100 mM NaCl, 1 M sucrose), the proteoliposomes were immediately fixed by the addition of glutaraldehyde and tannic acid at final concentrations of 2% and 0.05%, respectively. The fixed samples were collected by centrifugation, washed, post-fixed with 0.5% osmium tetroxide in 0.1 M phosphate buffer (pH 7.4), dehydrated, and embedded. Ultra-thin sections were stained with 2% uranyl acetate and further stained with lead stain solution (MilliporeSigma). Images of proteoliposomes were obtained using the transmission electron microscope JEM-1400Plus with the CCD camera VELETA. Fifty individual proteoliposomes with a diameter >500 nm were randomly selected, and proteoliposomes with invaginated membranes longer than one-third of the diameter were scored.

**Electron microscopy**. Yeast cells were sandwiched with the copper disks and frozen in liquid propane at −175 °C. After freeze substitution with 2% glutaraldehyde, 0.5% tannic acid, and 2% distilled water in acetone at −80 °C, the samples were warmed to 4 °C, rinsed with acetone, and fixed with 2% osmium tetroxide in acetone at 4 °C. After dehydration, the samples were infiltrated with propylene oxide, placed into a 70:30 mixture of propylene oxide and resin (Quetol-651; Nisshin EM Co), and embedded. Ultra-thin sections were stained with 2% uranyl acetate, washed, and further stained with lead stain solution (Sigma-Aldrich). Images of cells were obtained using a transmission electron microscope (JEM-1400Plus; JEOL Ltd.) with a CCD camera (VELETA; Olympus Soft Imaging Solutions GmbH).

For immune electron microscopy, HeLa cells transfected with plasmid carrying LETM1-3HA were sandwiched between molybdenum disks, frozen in liquid propane at −175 °C, and freeze-substituted with 0.025% osmium tetroxide and 3% distilled water. After embedding with LR-White resin (Sigma-Aldrich), ultra-thin sections were immunostained with anti-HA monoclonal antibody (Roche, clone 3F10) and gold-conjugated secondary antibody (British BioCell International), and further stained with lead stain solution. Images were collected using the transmission electron microscope JEM-1400Plus with the CCD camera VELETA.

**Expression and purification of His-LETM1 recombinant protein**. Expression and purification of a recombinant protein in silkworms were performed essentially as described previously[26]. For the expression of His-LETM1(107-739) recombinant protein, purified BmNPVbacmid DNA was mixed with the DMRIE-C reagent (Life Technologies) and then injected directly into fifth-instar silkworm larvae. Infected silkworm fat bodies were isolated, resuspended in PBS, and sonicated to disrupt the membranes of fat bodies. After ultracentrifugation at $100,000 \times g$ for 60 min, the resultant pellet was solubilized at 4 °C with 1% CHAPS (Dojindo Inc.) in buffer C [20 mM sodium phosphate (pH 7.4), 300 mM NaCl]. The lysate was further ultracentrifuged and the supernatant was applied to a HisTrap column (GE Healthcare). Purified recombinant proteins were dialyzed against buffer D [20 mM sodium phosphate (pH 7.4), 150 mM NaCl, 1% CHAPS].

**Statistics and reproducibility**. Statistical analysis was performed using two-tailed Student's $t$ test. In the column graphs, data represented the mean ± standard error of the mean (SEM). In the box and whisper plots, data represented the median with the interquartile ranges. $p$ values are described in the figure legends. All experiments were repeated three times with biological replicates to obtain the data analyzed.

**Reporting summary**. Further information on research design is available in the Nature Research Reporting Summary linked to this article.

## Data availability

All the data used are available from the corresponding author upon reasonable request. Source data are available in Supplementary Data 1.

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

## Acknowledgements

We greatly appreciate Atsumi Toyota for supporting the initial experiments of this study. We also thank Hiromi Hirose and Tomie Kameyama for excellent technical assistance. This work was supported by grants to T.O. from Grants-in-Aid for Scientific Research (17H03676) from JSPS, the Strategic Research Foundation Grant-aided Project for Private Universities (S1201003) from MEXT, and a JST CREST (Grant Number JPMJCR12M1).

## Author contributions

S.N., A.M., S.A., Y.T., and T.O. designed the experiments. S.N., A.M., S.A., Y.T., A.H., Y.Mi., H.O., M.K., and T.O. performed the experiments and analyzed the data. S.N., A.M., S.A., Y.T., Y.Mo., K.M., T.E., and T.O. contributed to preparing the manuscript.

## Competing interests

The authors declare no competing interests.
