## [Peer Review File · Communications Biology]

Reviewers' comments:

Reviewer #1 (Remarks to the Author):

Mammalian LETM1 and its yeast homolog Mdm38 are mitochondrial inner membrane proteins associated with various functions, including mitochondrial ion homeostasis, mitochondrial morphology, and mitochondrial protein translation. Here, Nakamura et al. propose that LETM1 is directly involved in the formation of cristae. While the study contains some interesting observations, it is rather preliminary and many conclusions are not sufficiently supported by the experiments, as outlined below.

Specific points

1. The authors argue that overexpression of LETM1-GFP induces mitochondrial fragmentation and that fragmentation depends on LETM1-GFP level (Fig. 1D). However, in Fig. 3C they show that LETM1-GFP is only marginally overexpressed in comparison to endogenous LETM1. How can this apparent contradiction be explained? The same is true for mitochondrial shrinkage by overexpression of LETM1-3HA. The authors should exclude the possibility that addition of a tag to the C terminus of LETM1 causes a dominant negative phenotype.
2. The electron microscopy shown in Fig. 1E is not clear. There is only a single mitochondrion with gold labeled cristae shown in panel b. More quantitative data are required. What are the dark organelles that are densely labeled in panel c? The phenotype of meshed inner membranes shown in panels d-f is striking. However, I understand that it was only observed after glutaraldehyde fixation. Therefore, the authors have to exclude that it is an artifact of the fixation. Non-transfected wild type cells have to be shown and the phenotypes have to be quantified in transfected and non-transfected cells.
3. The authors should indicate more clearly that MDM38 and LETM1 are strongly overexpressed in yeast. The information that the genes are expressed from the ADH1 promoter is hidden in Table 1. If very mild overexpression of LETM1 already causes severe phenotypes in mammalian cells (see above), wouldn't then much more severe phenotypes be expected upon strong overexpression in yeast cells?
4. The LETM1(D359A) allele supports growth of the yeast delta-mdm38 mutant on SCG at 30°C (at least in my eyes, this is more than a "slight support" of growth, as mentioned on p.8), however, it fails to support growth on YPG at 37°C. I wonder whether this is a temperature-sensitive phenotype, rather than an effect of the medium. This should be tested.
5. The authors should indicate medium and growth conditions used for the analysis of yeast mitochondrial morphology and ultrastructure in Fig. 2C and D. Furthermore, they should indicate which effects are statistically significant (Fig. 2C) and how many cells showed the mitochondrial cristae phenotypes (Fig. 2D).
6. Yeast mitochondrial cristae appear unusually long in Fig. 2D, panels a and c. Why do the membranes have negative contrast in panels a and c, but not in b, d, and e?
7. The authors conclude that missense mutations prevent correct assembly of LETM1 oligomers. However, the 250 kDa band is clearly visible in the CN PAGE in Fig. 3D. Why do the patterns of recombinant LETM1 in CN PAGE look much different from that of mitochondrial fractions (Fig. 3D versus 2B)?
8. It should read "Wolf-Hirschhorn syndrome" (Hirschhorn with o instead of the German umlaut ö).

Reviewer #2 (Remarks to the Author):

In this manuscript, authors report a direct role for Letm1 in the organization of cristae structure. They identify 4 amino acids within the LETM domain of LETM1 that seem to play a role in this cristae

organizing function and which are required to rescue growth deficiency caused by deletion of LETM1. In addition, authors demonstrate that recombinant LETM1 can cause invagination of membrane structures in an in vitro system of reconstituted proteoliposomes, suggesting a role for LETM1 in influencing membrane morphology. Although this is a potential interesting finding it is not clear as to why overexpression of LETM1 causes mitochondrial fragmentation and disruption of cristae and how this would consolidate with a proposed role for LETM1 in organizing cristae structure. The majority of the experiments, and thus the interpretations, are based on overexpression of LETM1 and thus it is not clear what the role of endogenous LETM1 would be. Given these points, and others stated below, a direct role for LETM1 in cristae organization per se has not been convincingly shown, making the title "Direct role of the mitochondrial inner membrane protein LETM1 in the organization of cristae structures" misleading.

Specific comments:

1) It is important for authors to show the effect of overexpression or depletion of LETM1 on actual cristae structure and organization. Although, authors nicely show that recombinant LETM1 can affect membrane morphology in an in vitro liposome system, this does not necessarily translate into a role for LETM1 in cristae organization. Performing EM analysis, measurements of cristae number and width, and detecting OPA1 oligomerization both in the presence and absence of LETM1 would provide a better correlation for a role in cristae organization.

2) It is not clear why overexpression of LETM1 would cause mitochondrial fragmentation. Would that not suggest that LETM1 plays a role in mitochondrial fission? Thus an alternate explanation to the role of LETM1 is that it may facilitate inner fragmentation during mitochondrial fission. This would in fact be a very important finding considering it remains unclear as to what mediates inner mitochondrial membrane fission within the context of mitochondrial dynamics.

3) Are the overexpression studies done in LETM1 KO or WT cells? Is it possible that the observed effect of LETM1 on mitochondrial structure (fragmentation and cristae disruption) may be due to toxicity related to overexpression of an exogenous protein. It is thus important to show the effects of knocking out LETM1 in these cells.

4) In addition to overexpressing LETM1, the authors should address the role of LETM1 by knockout or knockdown. Authors should investigate what happens to mitochondria in the absence of LETM1. Does it cause mitochondrial elongation? What happens to cristae structure under these conditions.

5) It is unclear as to why the authors switch between using Letm1-GFP in some studies, while using Letm1-3HA in others.

6) In this study the authors observe mitochondrial fragmentation and loss of cristae structure upon overexpression of LETM1 in cells. However, in an in vitro system, recombinant LETM1 results in membrane invagination. What is the link between the membrane invagination observed in vitro with the mitochondrial disruption observed in cells? The reviewer fails to conceive how these are connected and how this would lead to the interpretation that LETM1 organizes mitochondrial cristae.

7) Given the exception role of cristae in mitochondrial function and the direct role for cristae structure and OXPHOS, it is surprising that the authors have not tested the effect of LETM1 overexpression and deletion on mitochondrial function, especially those pertaining directly to cristae structure (ie. ATP generation, ROS levels ..etc). These data would strengthen the validity of a role for LETM1 in cristae organization/function.

8) The authors propose two models for how LETM1 make organize cristae, however no evidence for the first model is present in the manuscript. If LETM1 functioned as depicted in the first model then would you not have seen this in the in vitro liposome experiments where multiple invaginations would have been detected. Authors should perhaps further analyse the invaginations and quantify the number of invaginations per liposome.

9) How does 700kda band in Fig4B fit with the size of Letm1?? how many oligomers are forming?

10) In Fig3D, mutants show less assembly and authors state that "LETM1 missense mutations prevent the correct assembly of LETM1 oligomers". Does LETM1 only form oligomers with itself. How do you know that the mutants are not preventing the interaction of LETM1 with other LETM1-interacting proteins.

11) In Fig3C, authors state "the comparable expression level of the mutant proteins with that of the wild-type protein, however the western blot demonstrates quite different levels of expression of these proteins.

To Reviewers,

We were delighted to hear positive and kind comments from all of the reviewers. All comments and suggestions from the reviewers are very constructive and made our manuscript improved. We deeply appreciated your helpful cooperation.

Reviewer #1:

Mammalian LETM1 and its yeast homolog Mdm38 are mitochondrial inner membrane proteins associated with various functions, including mitochondrial ion homeostasis, mitochondrial morphology, and mitochondrial protein translation. Here, Nakamura et al. propose that LETM1 is directly involved in the formation of cristae. While the study contains some interesting observations, it is rather preliminary and many conclusions are not sufficiently supported by the experiments, as outlined below.

Specific points

1. The authors argue that overexpression of LETM1-GFP induces mitochondrial fragmentation and that fragmentation depends on LETM1-GFP level (Fig. 1D). However, in Fig. 3C they show that LETM1-GFP is only marginally overexpressed in comparison to endogenous LETM1. How can this apparent contradiction be explained? The same is true for mitochondrial shrinkage by overexpression of LETM1-3HA. The authors should exclude the possibility that addition of a tag to the C terminus of LETM1 causes a dominant negative phenotype.

We showed that moderate expression, but not overexpression, of LETM1-GFP caused mitochondrial fragmentation. To avoid a misunderstanding, we carefully rewrote the text of the revised manuscript without “overexpression”.

In the original manuscript, we showed that ectopic expression of non-tagged LETM1 also induced mitochondrial fragmentation (Supplementary Fig. 1A). In the revised manuscript, we described the following statements as “Similar results were obtained when non-tagged LETM1 protein was expressed ectopically (Supplementary Fig. 1A).” (Page 6, line 5~).

2. The electron microscopy shown in Fig. 1E is not clear. There is only a single mitochondrion with gold labeled cristae shown in panel b. More quantitative data are required.

According to the comment, we added the quantitative data of gold particle-positive submitochondrial membranes (Fig. 1F), and described the following statement as; “Sixty-five percent of gold particles in mitochondria were observed on the cristae membranes (Fig. 1F), indicating that LETM1 was localized in the cristae structures.” (Page 6, line 15~).

What are the dark organelles that are densely labeled in panel c? The phenotype of meshed inner membranes shown in panels d-f is striking. However, I understand that it was only observed after glutaraldehyde fixation. Therefore, the authors have to exclude that it is an artifact of the fixation. Non-

transfected wild type cells have to be shown and the phenotypes have to be quantified in transfected and non-transfected cells.

The cell shown in panel c of Fig. 1E was fixed as follows; cells were sandwiched between molybdenum disks, quickly frozen in liquid propane at -175°C, and freeze-substituted with 0.025% osmium tetroxide and 3% distilled water. Rapid freezing and freeze-substitution method is suitable for morphological preservation rather than chemical fixation (Giddings et al., *Methods Cell Biol.* 2001; **67**, 27-42). In the electron-dense organelles shown in panel c, the matrix was darker than the inside membranes, suggesting that matrix was condensed. Additionally, the electron-dense organelles were relatively smaller. Therefore, we assumed that electron-dense organelles shown in panel c are shrunken mitochondria, which is consistent with mitochondrial fragmentation caused by ectopic expression of LETM1. We showed the electron microscopic image of untransfected cells (Fig. 1E, panel a) in the original manuscript. To clearly describe it, we rewrote the figure legend and the text in the revised manuscript as follows; “(E) Untransfected cells (a) and cells transfected with the plasmid carrying LETM1-3HA (b, c) were fixed and subjected to immunoelectron microscopy using anti-HA antibody and gold particle-conjugated secondary antibody.” (Page 30, line 14~), and “Gold particles were clearly accumulated on cristae membranes of LETM1-3HA-expressing cells (Fig. 1E, panel b, arrowhead) compared with untransfected cells (panel a).” (Page 6, line 14). Moreover, we added the quantitative data of electron-dense mitochondria (Fig. 1I) and the corresponding statement as follows; “Such condensed mitochondria were frequently observed in LETM1-3HA-expressing cells compared with untransfected cells (Fig. 1I)” (Page 6, line 20~).

3. The authors should indicate more clearly that MDM38 and LETM1 are strongly overexpressed in yeast. The information that the genes are expressed from the ADH1 promoter is hidden in Table 1. If very mild overexpression of LETM1 already causes severe phenotypes in mammalian cells (see above), wouldn't then much more severe phenotypes be expected upon strong overexpression in yeast cells?

We apologize for incomplete description of the ADH promoter in Table 1 of the original manuscript. In the revised manuscript, we clearly described the information about the yeast expression plasmids and the promoter in Table 1. Yeast *MDM38* gene was expressed in yeast cells by the own promoter on a single-copy plasmid. In contrast, human LETM1 was expressed in yeast cells by yeast ADH1 promoter on a multicopy plasmid. Expression of human LETM1 on a single-copy plasmid was not enough to complement the growth defects of *Δmdm38* cells on a non-fermentable carbon source. Therefore, we used a multicopy plasmid for

complementation assay. Furthermore, unlike human cells, expression of human LETM1 on a multicopy plasmid in yeast cells did not cause aberrant mitochondrial morphology (Fig. 2C).

4. The *LETM1(D359A)* allele supports growth of the yeast *delta-mdm38* mutant on SCG at 30°C (at least in my eyes, this is more than a “slight support” of growth, as mentioned on p.8), however, it fails to support growth on YPG at 37°C. I wonder whether this is a temperature-sensitive phenotype, rather than an effect of the medium. This should be tested.

According to the reviewer’s suggestion, we examined the growth on SCG at 37°C of the yeast *mdm38* mutant cells expressing either wild type or mutants of human LETM1. Unfortunately, even when wild-type LETM1 was expressed, the growth defects of the yeast *mdm38* mutants on SCG at 37°C were not suppressed (See below, arrow). In contrast, wild-type LETM1 significantly supported the growth of the *mdm38* mutants on YPG at 37°C (Supplementary Fig. 3A). Thus, expression of human LETM1 partially suppresses the growth defects of the *mdm38* mutants, unlike the yeast *MDM38* gene.

5. The authors should indicate medium and growth conditions used for the analysis of yeast mitochondrial morphology and ultrastructure in Fig. 2C and D. Furthermore, they should indicate which effects are statistically significant (Fig. 2C) and how many cells showed the mitochondrial cristae phenotypes (Fig. 2D).

We apologize for incomplete description of medium and growth conditions for morphological analyses of the yeast strains. In the legend of Fig. 2, we clearly described the medium and growth conditions of the yeast strains as follows; “(C) Yeast *mdm38* mutant cells with the plasmid containing the indicated genes were transformed with a mitochondria-targeted GFP, and then cultured at 30°C onto SCD medium. Mitochondrial morphology of >50 individual cells was scored using live fluorescent images. Data represent the mean ± SEM of

three independent experiments. * $p < 0.005$; N. S., not significant ($p > 0.05$). (D) *mdm38* mutant cells carrying the indicated expression plasmids were cultured at 30°C onto SCD medium and mitochondrial membrane structures were analyzed by electron microscopy. Scale bar, 400 nm.” (Page 31, line 19~).

According to the comment, we added the quantitative data of yeast mitochondrial morphology (Fig. 2C) and ultrastructure (Supplementary Fig. 4C). Statistical analysis in Fig. 2C clearly showed that wild-type LETM1, but not LETM1-1 and LETM1-2 mutants, restored mitochondrial morphology in the yeast *mdm38* mutants. Additionally, quantitative analysis of ultrastructure (Supplementary Fig. 4C) showed that LETM1-1 and LETM1-2 mutants failed to suppress mitochondrial swelling in the yeast *mdm38* mutants.

6. *Yeast mitochondrial cristae appear unusually long in Fig. 2D, panels a and c. Why do the membranes have negative contrast in panels a and c, but not in b, d, and e?*

According to the comment, we improved electron microscopic images of mitochondria in the yeast *mdm38* mutants carrying yeast *MDM38* gene (panel a) and human LETM1 (panel c) in Fig. 2D. To easily compare between the yeast *mdm38* mutants and control wild-type strain in regard to changes in mitochondrial morphology, we added electron microscopic images of whole cell (Supplementary Fig. 4B). In control wild-type strain; i.e., the *mdm38* mutants carrying the *MDM38* gene, all mitochondria exhibited more electron dense than cytoplasm (panel a, yellow arrows). In contrast, mitochondria in the *mdm38* mutants carrying control vector were lighter than cytoplasm (panel b, blue arrows), due to mitochondrial swelling. Similarly, mitochondria of the yeast *mdm38* mutants carrying LETM1-1 or LETM1-2 mutants were lighter than cytoplasm (Fig. 2D, panels d, e) because LETM1-1 and LETM1-2 mutants could not restore mitochondrial swelling in the yeast *mdm38* mutants.

7. *The authors conclude that missense mutations prevent correct assembly of LETM1 oligomers. However, the 250 kDa band is clearly visible in the CN PAGE in Fig. 3D. Why do the patterns of recombinant LETM1 in CN PAGE look much different from that of mitochondrial fractions (Fig. 3D versus 2B)?*

LETM1 in mitochondria prepared from HeLa cells formed oligomers with the molecular size of 250 kDa on clear-native PAGE gels under our solubilizing conditions (Fig. 3D). Meanwhile, purified LETM1 recombinant proteins migrated as high-molecular and ladder-like bands on clear-native PAGE gels (Fig. 4B), suggesting that hyper-assembly of LETM1

oligomers in mitochondrial inner membranes is enzymatically repressed by regulatory proteins *in cell*. We reported that a mitochondrial inner membrane AAA-ATPase, BCS1L, regulates the assembly of LETM1 complexes (Tamai et al., *J. Cell Sci.* 2008, **121**, 2588-2600). Taken together, BCS1L may be involved in dissociation of highly-assembled LETM1 complexes *in cell*. As shown in Fig. 3D, LETM1-1 and LETM1-2 mutants formed smear high-molecular complexes (~1000 kDa) in addition to a 250-kDa oligomer. In contrast, most of wild-type LETM1 formed a 250-kDa oligomer under the same condition. As described above, hyper-assembly of LETM1 complexes (~1000 kDa) is usually repressed *in cell*. These results suggest that LETM1-1 and LETM1-2 mutants form incorrectly-assembled complexes that may be insensitive to disassembly mediated by regulatory proteins including BCS1L. In the discussion section of revised manuscript, we described as follows; “Endogenous LETM1 protein oligomerizes with apparent molecular sizes of 250 kDa ~ 500 kDa. Purified His-LETM1 recombinant proteins tended to migrate ladder-like bands on clear-native PAGE gels *in vitro* and the complexes of His-LETM1 proteins were larger than those of endogenous LETM1 proteins observed *in cell*, suggesting that hyper-assembly of LETM1 complexes in mitochondrial inner membranes is enzymatically and spatially repressed by other proteins and/or lipids *in cell*. Indeed, a mitochondrial inner membrane AAA-ATPase, BCS1L, regulates the assembly of LETM1 complexes. Both LETM1 mutant proteins prepared from HeLa cells formed large and various sizes of complexes (250 kDa -1000 kDa), suggesting that large complexes of LETM1 mutant proteins may be insensitive to disassembly mediated by regulatory proteins.” (Page 15, line 13~).

8. It should read “Wolf-Hirschhorn syndrome” (Hirschhorn with o instead of the German umlaut ö).

We corrected all statements about “Wolf-Hirschhorn syndrome” in the revised manuscript.

Reviewer #2:

In this manuscript, authors report a direct role for Letm1 in the organization of cristae structure. They identify 4 amino acids within the LETM domain of LETM1 that seem to play a role in this cristae organizing function and which are required to rescue growth deficiency caused by deletion of LETM1. In addition, authors demonstrate that recombinant LETM1 can cause invagination of membrane structures in an in vitro system of reconstituted proteoliposomes, suggesting a role for LETM1 in influencing membrane morphology. Although this is a potential interesting finding it is not clear as to why overexpression of LETM1 causes mitochondrial fragmentation and disruption of cristae and how this would consolidate with a proposed role for LETM1 in organizing cristae structure. The majority of the experiments, and thus the interpretations, are based on overexpression of LETM1 and thus it is not clear what the role of endogenous LETM1 would be. Given these points, and others stated below, a direct role for LETM1 in cristae organization per se has not been convincingly shown, making the title "Direct role of the mitochondrial inner membrane protein LETM1 in the organization of cristae structures" misleading.

According to the comment, we discussed a relation of LETM1 upregulation to mitochondrial fragmentation and aberrant cristae structures in the revised manuscript (See the response to the comment #2 and #8).

In the original manuscript, we discussed, in addition to our current findings with regards to LETM1 upregulation, about the results of LETM1 knockdown that were previously reported by our group and other researchers (Tamai *et al.*, *J. Cell Sci.* 2008, **121**, 2588-2600; Dimmner *et al.*, *Hum. Mol. Gent.* 2007, **17**, 201-214; Hasegawa and van der Bliek, *Hum. Mol. Gent.* 2007, **17**, 2061-2071). Combined with the results of upregulation and downregulation of LETM1 expression, we made the title of the original manuscript. As suggested by the reviewer, we carefully thought a title and removed "Direct" from a title of the revised manuscript.

Specific comments:

1) It is important for authors to show the effect of overexpression or depletion of LETM1 on actual cristae structure and organization. Although, authors nicely show that recombinant LETM1 can affect membrane morphology in an in vitro liposome system, this does not necessarily translate into a role for LETM1 in cristae organization. Performing EM analysis, measurements of cristae number and width, and detecting OPA1 oligomerization both in the presence and absence of LETM1 would provide a better correlation for a role in cristae organization.

According to the comment, we analyzed quantitatively the cristae junction and width of the LETM1-expressing cells using electron microscopic images. We did not count correctly the number of cristae membranes, because mitochondria in the LETM1-expressing cells exhibited the meshed membrane structures. Therefore, we counted cristae junctions, instead of cristae membranes. Upon the ectopic expression of LETM1, the numbers of cristae junctions

remained unchanged (Fig. 1H) and the mean width of cristae structures increased approximately 1.7-fold (Fig. 1J). These results suggested that the meshed membrane structures are formed by the adhesion of two adjacent invaginated membranes. Furthermore, we examined OPA1 oligomerization upon both knockdown and upregulation of LETM1, as suggested by the reviewer. As expected, the formation of OPA1 oligomers decreased when LETM1 was knocked down (Supplementary Fig. 1F), because LETM1 knockdown causes mitochondrial swelling and disruption of cristae structures (Tamai *et al.*, *J. Cell Sci.* 2008, **121**, 2588-2600). In contrast, the ectopic expression of LETM1 did not affect OPA1 oligomerization (Supplementary Fig. 1E), which is consistent with the absence of an influence on the number of cristae junctions. And the results supported the idea that the increase in width of the cristae membranes was due to the adhesion of two invaginated membranes. We added these four figures to the revised manuscript and described the corresponding statements as follows; “The numbers of cristae junctions remained unchanged in the LETM1-3HA-expressing cells (Fig. 1H), suggesting that formation of the meshed inner membranes was not due to changes in the number of the invaginated membranes. A high magnification image showed an increase in the width of the meshed membranes (Fig. 1G, panel d), compared with cristae membranes in untransfected cells (panel a). In LETM1-3HA-expressing cells, the mean cristae width increased approximately 1.7-fold (Fig. 1J). Taken together with the absence of an influence on the number of cristae junctions, these results suggest that the meshed membrane structures are formed by the adhesion of two adjacent invaginated membranes. Formation of OPA1 oligomers is linked to changes in cristae structures and OPA1 oligomerization is stimulated in mitochondria with narrow cristae structures. We examined the effects of LETM1 ectopic expression on the formation of OPA1 oligomers using a membrane-permeable cross linker. As expected, LETM1 knockdown induced a reduction in OPA1 oligomer formation (Supplementary Fig. 1F), because LETM1 downregulation causes mitochondrial swelling and disruption of cristae structures. Ectopic expression of LETM1, however, did not affect OPA1 oligomerization (Supplementary Fig. 1E), supporting the idea that the increased cristae width in the meshed membranes was due to the adhesion of the invaginated membranes, and not to dilated cristae structures.” (Page 7, line 7~).

2) *It is not clear why overexpression of LETM1 would cause mitochondrial fragmentation. Would that not suggest that LETM1 plays a role in mitochondrial fission? Thus an alternate explanation to the role of*

LETM1 is that it may facilitate inner fragmentation during mitochondrial fission. This would in fact be a very important finding considering it remains unclear as to what mediates inner mitochondrial membrane fission within the context of mitochondrial dynamics.

We greatly appreciated the reviewer's suggestion. According to the suggestion, we discussed the role of LETM1 in mitochondrial fission in the revised manuscript. Cytoplasmic GTPase Drp1 mainly contributes to mitochondrial fission and its knockdown induces mitochondrial elongation. In contrast, LETM1 knockdown causes mitochondrial swelling. Double knockdown of Drp1 and LETM1 caused both elongation and swelling of mitochondrial filaments (Tamai *et al.*, *J. Cell Sci.* 2008, **121**, 2588-2600; Dimmner *et al.*, *Hum. Mol. Genet.* 2007, **17**, 201-214), suggesting that LETM1 contributes to mitochondrial morphology independently of the Drp1-mediated membrane fission. Furthermore, LETM1 upregulation caused mitochondrial stress, including ROS generation and the low membrane potential, suggesting that mitochondrial fragmentation caused by LETM1 upregulation may be indirectly stimulated by mitochondrial stress. We added the following statements in the discussion section; "Exogenous expression of LETM1 induced mitochondrial fragmentation. Mitochondrial fission is governed by cytoplasmic GTPase Drp1. Drp1 knockdown stimulates elongation of mitochondrial filaments. In contrast, LETM1 knockdown causes large dot-like structures of mitochondria. Simultaneous knockdown of Drp1 and LETM1 results in an additive phenocopy; i.e., both elongation and swelling of mitochondrial filaments, suggesting that LETM1 functions in mitochondrial morphology separately from the Drp1-mediated membrane fission pathway. Furthermore, LETM1 knockdown induces a reduction in mitochondrial membrane potential and exogenous expression of LETM1 caused mitochondrial damage, including ROS generation and the low membrane potential, suggesting that changes in LETM1 expression may induce mitochondrial stress through morphological alterations of mitochondrial inner membranes. Therefore, mitochondrial fragmentation caused by LETM1 upregulation may be indirectly stimulated by mitochondrial stress." (Page 15, line 1~).

3) Are the overexpression studies done in LETM1 KO or WT cells? Is it possible that the observed effect of LETM1 on mitochondrial structure (fragmentation and cristae disruption) may be due to toxicity related to overexpression of an exogenous protein. It is thus important to show the effects of knocking out LETM1 in these cells.

We did not use the LETM1-knockout cells for any experiments. Because homozygous deletion of *letm1* in mice caused embryonic lethal and half of the heterozygotes also died before birth (Jiang *et al.*, *PNAS* 2013, **110**, E2249-E2254). In addition, fibroblasts

derived from survived human patients with Wolf-Hirschhorn syndrome showed normal mitochondria morphology (Dimmner *et al.*, *Hum. Mol. Gent.* 2007, **17**, 201-21), suggesting that constitutive gene silencing of LETM1 induces cellular adaptation during cell survival. Ectopic expression of LETM1-1 and LETM1-2 mutants barely influenced mitochondrial morphology (Fig. 3A, 3B), suggesting that changes in morphology including mitochondrial fragmentation and aberrant cristae structures are related to the LETM1 functions but are not due to exogenous expression of mitochondrial proteins.

4) *In addition to overexpressing LETM1, the authors should address the role of LETM1 by knockout or knockdown. Authors should investigate what happens to mitochondria in the absence of LETM1. Does it cause mitochondrial elongation? What happens to cristae structure under these conditions.*

Our group and other researchers have already reported that LETM1 knockdown causes mitochondrial swelling and disruption of cristae structures (Tamai *et al.*, *J. Cell Sci.* 2008, **121**, 2588-2600; Dimmner *et al.*, *Hum. Mol. Gent.* 2007, **17**, 201-214; Hasegawa and van der Blik, *Hum. Mol. Gent.* 2007, **17**, 2061-2071). Electron microscopic analysis of the LETM1-knockdowned cells showed that investigated membrane structures were barely observed in swollen mitochondria (Tamai *et al.*, *J. Cell Sci.* 2008, **121**, 2588-2600). We described about the phenocopy of LETM1 knockdown in the revised manuscript as follows; “Knockdown of LETM1 induces mitochondrial swelling and the disappearance of cristae structures.” (Page 5, line 19~), “LETM1 knockdown causes large dot-like structures of mitochondria.” (page 15, line 3~), and “LETM1 knockdown causes the disappearance of the cristae structures and mitochondrial swelling.” (Page 17, line 10~).

5) *It is unclear as to why the authors switch between using Letm1-GFP in some studies, while using Letm1-3HA in others.*

We used a plasmid coexpressing LETM1-3HA and a nuclear targeted GFP under a single promoter for first analysis of live imaging. Because GFP intensity is proportionally related to the LETM1 expression level. In Supplementary Fig. 1A, we showed fluorescent microscopic images of cells carrying a plasmid coexpressing non-tagged LETM1 and a nuclear targeted GFP. However, the intensity of GFP-fused proteins directly represents the expression levels, which helps the readers to easily understand a relation of the increased LETM1

expression to changes in mitochondrial morphology. Therefore, we decided to use a plasmid expressing LETM1-GFP in Fig. 1D and 3A.

6) in this study the authors observe mitochondrial fragmentation and loss of cristae structure upon overexpression of LETM1 in cells. However, in an in vitro system, recombinant LETM1 results in membrane invagination. What is the link between the membrane invagination observed in vitro with the mitochondrial disruption observed in cells? The reviewer fails to conceive how these are connected and how this would lead to the interpretation that LETM1 organizes mitochondrial cristae.

Knockdown of LETM1 causes the disappearance of the cristae structures.

Upregulation of LETM1 led to formation of the meshed cristae membranes, but *NOT* loss of cristae structures. These results suggested that LETM1 facilitates adhesion of adjacent inner membranes *in cell*. Moreover, LETM1 recombinant proteins formed *in vitro* membrane invagination, indicating that LETM1 proteins can facilitate adhesion of nearest membranes within single liposomes. Thus, LETM1 may mediate the formation of appropriate adhesion of mitochondrial inner membranes to maintain cristae structures. To clearly describe the possibility how LETM1 organizes cristae structures, we rewrote the corresponding statements in the discussion section as follows; “LETM1 knockdown causes the disappearance of the cristae structures and mitochondrial swelling. Furthermore, upregulation of LETM1 led to formation of the meshed cristae membranes and mitochondrial shrinkage, suggesting that LETM1 facilitates adhesion of neighboring inner membranes. Additionally, LETM1 recombinant proteins formed invaginated membranes *in vitro*, indicating that LETM1 proteins can facilitate the adherence of nearest parts of membranes within single artificial liposomes. Thus, LETM1 may mediate the formation of appropriate adhesion of mitochondrial inner membranes to maintain cristae structures.” (Page 17, line 10).

7) Given the exception role of cristae in mitochondrial function and the direct role for cristae structure and OXPHOS, it is surprising that the authors have not tested the effect of LETM1 overexpression and deletion on mitochondrial function, especially those pertaining directly to cristae structure (ie. ATP generation, ROS levels ..etc). These data would strengthen the validity of a role for LETM1 in cristae organization/function.

According to the reviewer’s suggestions, we examined the effect of LETM1 upregulation on mitochondrial ROS levels. Thirty-three percent of cells expressing LETM1-GFP were clearly stained with MitoSOX Red, a membrane-permeable indicator for monitoring mitochondrial ROS, whereas GFP-transfected cells were barely stained (Supplementary Fig.

2B). In addition, ectopic expression of LETM1 induced the low membrane potential (Supplementary Fig. 2B), and LETM1 knockdown also causes a reduction in mitochondrial membrane potential (Tamai *et al.*, *J. Cell Sci.* 2008, **121**, 2588-2600). These results supported that LETM1 plays a role in the organization and function of cristae structures. We added the following statements as; “To examine the effect of exogenous LETM1 expression on the membrane potential, cells transfected with LETM1-GFP were stained with the membrane potential-dependent fluorescent dye, tetramethylrhodamine methyl ester (TMRM). The LETM1-GFP-expressing cells were barely stained with TMRM compared with untransfected cells (Supplementary Fig. 2A), indicating that cells expressing exogenous LETM1 had low mitochondrial membrane potentials. Production of mitochondrial reactive oxygen species (ROS) is affected by changes in the membrane potential. To test whether the ectopic expression of LETM1 induces ROS production, LETM1-GFP-transfected cells were stained with the fluorescent dye MitoSOX Red, a membrane-permeable indicator for monitoring mitochondrial ROS. Thirty-three percent of cells expressing LETM1-GFP were clearly stained with MitoSOX Red (Supplementary Fig. 2B), indicating that exogenous expression of LETM1 led to increased ROS production.” Page 8, line 4~).

8) *The authors propose two models for how LETM1 make organize cristae, however no evidence for the first model is present in the manuscript. If LETM1 functioned as depicted in the first model then would you not have seen this in the in vitro liposome experiments where multiple invaginations would have been detected. Authors should perhaps further analyse the invaginations and quantify the number of invaginations per liposome.*

So far, we could not observe giant proteoliposomes (diameter: more than 500 nm) with multiple membrane invaginations. In our *in vitro* assay, we counted giant proteoliposomes with single invaginated membranes that was longer than one-third of the diameter. Because of size limitation (diameter: less than 1 μ m) of proteoliposome prepared in our assay, multiple membrane invaginations might be not formed. According to the reviewer’s suggestion, we removed the cartoon of two models and the corresponding sentences in the discussion section.

9) *How does 700kda band in Fig4B fit with the size of Letm1?? how many oligomers are forming?*

As mentioned above (response to the comment #7 of the reviewer #1), LETM1 usually forms complexes with molecular weights ranging from 250 kDa to 500 kDa on native gels (Tamai *et al.*, *J. Cell Sci.* 2008, **121**, 2588-2600; Dimmner *et al.*, *Hum. Mol. Gent.* 2007,

17, 201-214; Hasegawa and van der Bliek, *Hum. Mol. Gent.* 2007, **17**, 2061-2071). The size heterogeneity is dependent on the electrophoretic conditions including detergents and Coomassie dyes (Supplementary Fig. 5A). Purified wild-type LETM1 recombinant proteins migrated as ladder bands with molecular sizes ranging from 500 kDa to 1100 kDa on clear-native PAGE gels (Fig. 4B). *In vitro* oligomers of LETM1 recombinant proteins were larger than the complexes of LETM1 proteins observed *in cell*, suggesting that that hyper-assembly of LETM1 in mitochondrial inner membranes is enzymatically suppressed by other proteins *in cell*. We reported that a mitochondrial inner membrane AAA-ATPase, BCS1L, regulates the assembly of LETM1 complexes (Tamai et al., *J. Cell Sci.* 2008, **121**, 2588-2600). Taken together, BCS1L may regulate appropriate assembly of LETM1 complexes *in cell*.

10) *In Fig3D, mutants show less assembly and authors state that “LETM1 missense mutations prevent the correct assembly of LETM1 oligomers”. Does LETM1 only form oligomers with itself. How do you know that the mutants are not preventing the interaction of LETM1 with other LETM1-interacting proteins.*

According to the comment, we rewrote the corresponding sentence as follows:

“whereas the LETM1-1 and LETM1-2 mutant proteins failed to migrate as only a single band and formed smeared bands with apparent molecular weights ranging from 250 kDa to more than 1000 kDa (Fig. 3D), suggesting that LETM1 missense mutations repress the correct assembly of LETM1-containing protein complexes.” (Page 11, line 9~; changes are underlined).

11) *In Fig3C, authors state “the comparable expression level of the mutant proteins with that of the wild-type protein, however the western blot demonstrates quite different levels of expression of these proteins.*

According to the comment, we reexamined immunoblotting of wild type and mutant LETM1-GFP proteins. Semi-quantitative analysis of the immunoblotting showed the comparable expression level of both LETM1-1 and LETM1-2 mutant proteins with that of the wild-type protein. The ratio of wild-type LETM1-GFP to mutant LETM1-GFP proteins was as follows; 100% (wild type) : 93% (LETM1-1) : 83% (LETM1-2). New figure was incorporated as Fig. 3C in the revised manuscript.

Reviewers' comments:

Reviewer #1 (Remarks to the Author):

The revisions improved the manuscript. However, the following points still need to be addressed (the numbers refer to my original report):

1. The expression levels of LETM1-GFP are still not clear to me. In the original manuscript the authors wrote: "Expression of LETM1-GFP, however, was 79% lower than that of endogenous LETM1 (Fig. 3C)"; in the revised manuscript they write: "Expression of LETM1-GFP was comparable to that of endogenous LETM1 (Fig. 3C)." Which is correct? I would have appreciated an explanation of this discrepancy, at least in the rebuttal letter.
2. The authors now state that "Sixty-five percent of gold particles in mitochondria were observed on the cristae membranes (Fig. 1F), indicating that LETM1 was localized in the cristae structures." They added a quantification in Fig. 1F. Technically it is not possible to discriminate a localization in the outer membrane versus inner boundary membrane, or in cristae versus matrix (due to the size of the antibody the gold particle does not exactly localize to the site of the antigen). Therefore, the quantification should discriminate between the rim of the organelle and its interior. Importantly, the authors should also take into account that there is much more cristae membrane than inner boundary membrane. Therefore, they have to calculate the number of gold particles per nm of membrane. The authors have added electron micrographs to Fig. S1D. These are exactly the same cells that are already shown in Fig. 1Ga and 1Gc and d. I don't see the point of showing the same data twice. How many biological replicates were analyzed for the EM analysis? In their rebuttal letter, the authors cite Giddings et al. 2001 for their EM protocol. This reference should be added to the manuscript.
3. The yeast experiments are still not entirely clear. For example, in Fig. 2B strains are labelled "Wild type Vector", "deltamd38 Yeast MDM38" etc.; however, in Table 1 the designation of the plasmids is different. The source of most of the plasmids listed in Fig. 1 is indicated as "This study"; unfortunately, there is no description of the cloning of these plasmids.

Reviewer #2 (Remarks to the Author):

The authors have made substantial revisions and added important experiments that strengthen the conclusions of the study. I endorse this manuscript for publication.

Reviewer #1

The revisions improved the manuscript. However, the following points still need to be addressed (the numbers refer to my original report):

We greatly appreciated your constructive comments and helpful suggestions, which made our manuscript improved.

1. The expression levels of LETM1-GFP are still not clear to me. In the original manuscript the authors wrote: "Expression of LETM1-GFP, however, was 79% lower than that of endogenous LETM1 (Fig. 3C)"; in the revised manuscript they write: "Expression of LETM1-GFP was comparable to that of endogenous LETM1 (Fig. 3C)." Which is correct? I would have appreciated an explanation of this discrepancy, at least in the rebuttal letter.

We apologized for the incomplete description about LETM1-GFP expression in our last response letter. For revision of the manuscript, we repeated the experiment to quantify LETM1-GFP expression several times. The expression of LETM1-GFP varied in the experiments because it appeared to be affected by DNA transfection condition and preparation of the expression plasmids carrying LETM1-GFP. However, the expression levels of LETM1-GFP were not higher than that of endogenous LETM1 at any time. In Fig. 3C, it was necessary to show that LETM1-GFP was not overexpressed when compared to endogenous LETM1. We thought that it was inappropriate to emphasize lower expression of LETM1-GFP. Therefore, we chose the data showing the comparable expression of LETM-GFP to endogenous LETM1. Because it was enough to indicate that LETM1-GFP was not overexpressed even when the data of the comparable expression of LETM-GFP were used.

2. The authors now state that "Sixty-five percent of gold particles in mitochondria were observed on the cristae membranes (Fig. 1F), indicating that LETM1 was localized in the cristae structures." They added a quantification in Fig. 1F. Technically it is not possible to discriminate a localization in the outer membrane versus inner boundary membrane, or in cristae versus matrix (due to the size of the antibody the gold particle does not exactly localize to the site of the antigen). Therefore, the quantification should discriminate between the rim of the organelle and its interior. Importantly, the authors should also take into account that there is much more cristae membrane than inner boundary membrane. Therefore, they have to calculate the number of gold particles per nm of membrane.

We greatly appreciated the constructive comment that made our manuscript improved. We agreed that it is difficult technically to discriminate a gold particle's localization in the outer membrane/inner boundary membrane, due to the size of the antibody. According to the reviewer's suggestion, we planned to use the method for calculating the number of gold particles per nm of membrane, which was reported in the previous paper using yeast cells (Vogel et al, *J. Cell Biol.* 2006, **175**, 237-247). However, the expression of exogenous LETM1 induced mitochondrial shrink. As a consequence, the lengths of mitochondrial membranes varied and appeared to depend on the expression levels of exogenous LETM1. Therefore, we

counted the number of gold particles on the rim and the interior of the mitochondria because it was enough to support that LETM1 was an inner membrane protein that was mainly localized in the cristae membrane. We added a new figure (Fig. 1F) and the following statement as; “Most of gold particles in mitochondria were observed on the interior of mitochondria (Fig. 1F), suggesting that LETM1 was an inner membrane protein localized mainly in the cristae membranes.” (page 6, line 17~)

The authors have added electron micrographs to Fig. 1D. These are exactly the same cells that are already shown in Fig. 1Ga and 1Gc and d. I don't see the point of showing the same data twice. How many biological replicates were analyzed for the EM analysis?

We thought that high magnification images of mitochondria of LETM1-3HA-expressing cells (Fig. 1G) may be insufficient to show that condensed and shrunk mitochondria were not due to an artifact of the fixation. To support that cellular structures were well preserved by our method, we added the low magnification images of both control and LETM1-3HA-expressing cells to Supplementary figures. As suggested by the reviewer's comment, we removed the low magnification images (Supplementary Fig. 1D in the last manuscript). Furthermore, we added the statement to figure legend as follows; “(H-J) Cristae junctions per mitochondrion (H), electron-dense mitochondria per cell (I), and cristae width (J) of the sections described in (G) were quantified. Centerlines, boxes, and whiskers represent the median, the interquartile range, and the full extent of the data of three independent experiments, respectively; >20 individual mitochondria were counted.” (page 31, line 21~)

In their rebuttal letter, the authors cite Giddings et al. 2001 for their EM protocol. This reference should be added to the manuscript.

According to the comment, we cited the paper and added the statement in text as follows; “To investigate the membrane structures of mitochondria of cells expressing LETM1-3HA, we performed immunoelectron microscopy by rapid freezing and freeze-substitution method, which is suitable for morphological preservation rather than chemical fixation (Giddings et al., *Methods Cell Biol.* 2001; **67**, 27-42).” (page 6, line 10~)

3. The yeast experiments are still not entirely clear. For example, in Fig. 2B strains are labelled “Wild type Vector”, “deltamd38 Yeast MDM38” etc.; however, in Table 1 the designation of the plasmids is different. The source of most of the plasmids listed in Fig. 1 is indicated as “This study”; unfortunately, there is no description of the cloning of these plasmids.

We thank for the helpful comment. According to the comment, we improved Fig. 2B

and Supplementary Fig. 3A. We used “yeast strain” and “expressed protein”, which were explained in Table 2 and Table 1, respectively. Furthermore, we described the construction of the plasmids in Methods as follows; “To construct expression plasmids for LETM1-GFP, a DNA fragment containing LETM1 was digested from pEF1-LETM1-3HA (Tamai et al., *J. Cell Sci.* 2008, **121**, 2588-2600), and inserted into pAcGFP-N1 (Clontech). LETM1-1 and LETM1-2 mutants were constructed by PCR using the primers (LETM1-1, 5'-ggcagcagcagcagcgggcccctgggcgtcacggaa-3' and 5'-gcccgctgctgctgctgcccgacacgctgcctgcag-3'; LETM1-2, 5'-ggcagacgctaagctgattgctgaggaaggggtg-3' and 5'-aatcagcttagcgtctgcctttatggagcgcagc-3') and cloned into pEF1-3HA and pAcGFP-N1. For the expression of human LETM1 in yeast cells, the ADH1 promoter and CYC1 terminator of pMID-2 (Horie et al., *J. Biol. Chem.* 2003, **278**, 41462-41471) were subcloned into a yeast multicopy vector pRS426 to construct pADH1. A DNA fragment of LETM1-3HA from pEF1-LETM1-3HA was inserted into pADH1 to construct pADH1-LETM1-3HA. A 2.5-kb DNA fragment containing the yeast MDM38 gene and its promoter region was amplified by PCR from a yeast genomic DNA using the primers (5'-acttgtgcagagccttgatcc-3' and 5'-aaaaggtaccagctttactgcgtgcattac-3') and cloned into a single-copy vector pRS316 to construct pRS316-MDM38.” (page 18, line 19~).

Reviewer #2

The authors have made substantial revisions and added important experiments that strengthen the conclusions of the study. I endorse this manuscript for publication.

We were delighted by your favorable response to our revised manuscript and deeply appreciated your constructive comments and helpful suggestions.